# Rule Extrapolation in Language Models: A Study of Compositional Generalization on OOD Prompts

**Anna Mészáros**[1], **Szilvia Ujváry**[1,5], **Wieland Brendel**[2,3,4], **Patrik Reizinger**[*2], and **Ferenc Huszár**[*1]

[1]University of Cambridge, Cambridge, United Kingdom
[2]Max Planck Institute for Intelligent Systems, Tübingen, Germany
[3]ELLIS Institute Tübingen, Tübingen, Germany
[4]Tübingen AI Center, Tübingen, Germany
[5] AI Center, UCL, London, United Kingdom

## Abstract

LLMs show remarkable emergent abilities, such as inferring concepts from presumably out-of-distribution prompts, known as in-context learning. Though this success is often attributed to the Transformer architecture, our systematic understanding is limited. In complex real-world data sets, even defining what is out-of-distribution is not obvious. To better understand the OOD behaviour of autoregressive LLMs, we focus on formal languages, which are defined by the intersection of rules. We define a new scenario of OOD compositional generalization, termed *rule extrapolation*. Rule extrapolation describes OOD scenarios, where the prompt violates at least one rule. We evaluate rule extrapolation in formal languages with varying complexity in linear and recurrent architectures, the Transformer, and state space models to understand the architectures' influence on rule extrapolation. We also lay the first stones of a normative theory of rule extrapolation, inspired by the Solomonoff prior in algorithmic information theory.

## 1   Introduction

Autoregressive language models (AR LMs) can reach both low training and test loss, but even minimal test loss is not predictive for out-of-distribution (OOD) model performance [Liu et al., 2023, Reizinger et al., 2024], i.e. when the test data has vanishing probability under the training distribution. Despite the success of deploying modern language models in OOD situations, OOD generalization is not well understood theoretically. Recently, studies started to focus on a specific form of OOD generalization: compositional generalization in language models [Ahuja and Mansouri, 2024, Han and Padó, 2024, Ramesh et al., 2024, Lake and Baroni, 2023, Reizinger et al., 2024]. To systematically examine compositional generalization of AR LMs, we study a particular notion of OOD generalization, which we call rule extrapolation.

*Rule extrapolation is a form of compositional generalization: it studies OOD behavior of language models trained on formal languages defined by a logical conjunction of rules.*

For example, the $a^n b^n$ language is the intersection of two rules: (R1) the number of $a$'s is equal to the number of $b$'s and (R2) $a$'s precede $b$'s. The prompt **bbaab** cannot be completed to obey the R2, but it is still possible to satisfy (R1) (e.g., **bbaab**$a$). When a language model trained on an intersection of rules remains consistent with one of the rules when another is broken, we say it successfully extrapolated the rule beyond its training data.

---

*Joint senior authors. Correspondence to `am3049@cam.ac.uk`. Code available at: `github.com/meszarosanna/rule_extrapolation`

A limited experiment by Reizinger et al. [2024] indicated that Transformers exhibit much-better-than-chance rule extrapolation performance on the formal grammar $a^n b^n$, despite lacking any explicit inductive biases encouraging this behaviour. However, it remains unclear whether the behaviour observed was specific to the Transformer or whether it holds more generally on a wider range of formal languages. Inspired by this work, we conduct a thorough empirical investigation of the role of architecture in rule extrapolation on a range of formal languages. As a non-rigorous baseline, we also conducted a small pilot human study to understand how people would generalize the rules.

We chose to study rule extrapolation because it appears to be a rational, or at least desirable, behaviour. However, we lack a normative reason why this behaviour should be considered rational. It is unclear whether any OOD behaviours could be considered rational. This question led us to investigate how a general rational algorithm for OOD prompt completion might be formalized. That is, instead of asking what models do, we ask what they *should* do if they were to be consistent with some principles of rational inference. We turn to Algorithmic Information Theory (AIT) to formalize a normative model. We propose a non-parametric prior for next-token prediction inspired by the Solomonoff prior [Solomonoff, 2001, Li and Vitányi, 1997]. This prior helps resolve how a rational model should behave in situations that are mathematically underspecified by their training: to extrapolate the simplest theories consistent with training data. Although, like Solomonoff's induction, our rational algorithm is uncomputable, it helps explain some of our empirical observations about rule extrapolation in practical language models. Our **contributions** are:

- We use formal languages to define scenarios for evaluating sequence models' OOD compositional generalization, which we call *rule extrapolation* (§ 2.2);
- We empirically evaluate different models' rule extrapolation in formal languages with varying complexity, we study linear, recurrent, Transformer and State Space models. We show that there is no single architecture that emerges as a clear winner of rule extrapolation. Though Transformers fare very well in most scenarios we investigated, they struggle on regular languages (§ 4);
- Inspired by algorithmic information theory, we propose a normative theory for OOD prompt completion, which posits that rule learning and extrapolation should be governed by the relative simplicities of rules (§ 5);
- To demonstrate the presence of a similar simplicity bias in Transformers, We visualise the training dynamics enabling rule extrapolation on the $a^n b^n$ language. We find that the model first learns a set obeying the easier rule, and then identifies the language as its subset (§ 5.3).

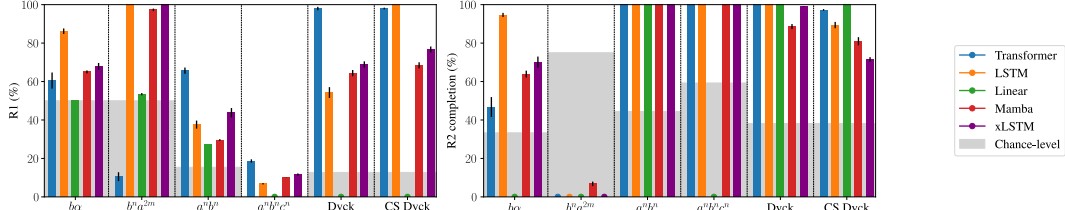

Figure 1: **Rule extrapolation summary for all models and languages (Tab. 1):** The Transformer is the best on context-free and context-sensitive languages, whereas the LSTM and Mamba excel on regular languages. We also plot chance-level performance as gray rectangles. Mean accuracies and standard deviations (averaged over 5 seeds)

| Language | Category | Rule 1 | Rule 2 |
|---|---|---|---|
| $L_1 = \{b\alpha\}$ | regular | #$a$ even | starts with $b$ |
| $L_2 = \{b^n a^{2m}\}$ | regular | #$a$ even | $b$'s before $a$'s |
| $L_3 = \{a^n b^n\}$ | context-free | #$a$ = #$b$ | $a$'s before $b$'s |
| $L_4 = $ Dyck | context-free | paired and nested $[\,]$ | paired and nested $(\,)$ |
| $L_5 = \{a^n b^n c^n\}$ | context-sensitive | #$a$ = #$b$ = #$c$ | $a$'s before $b$'s before $c$'s |
| $L_6 = $ CS Dyck | context-sensitive | paired $[\,]$ | paired $(\,)$ |

Table 1: **Formal languages used in our paper:** The languages are categorized according to the Chomsky hierarchy, and they can be considered as the intersection of two rules: (R1) and (R2)

## 2 Background and related work

### 2.1 Formal languages

Formal languages are linguistic constructions that simplify the study of natural languages. Their advantage is their well-defined set of symbols and rules. Although they fall short of capturing the nuances and irregularities of human languages, they are very powerful with immense practical relevance—e.g., programming languages are formal languages.

Formal languages consist of words with symbols coming from a possibly infinite alphabet. Chomsky [1956] has categorized formal languages into four types with increasing complexity: regular, context-free, context-sensitive, and recursively enumerable languages. *Regular* languages have rules that can be expressed via regular expressions, e.g., $L_1 = \{b\alpha : \alpha \text{ contains even number of } a\text{'s}\}$ and $L_2 = \{b^n a^{2m} : n > 0\}$. *Context-free grammars* have rules that do not depend on the context—programming languages such as C or Python belong to this category, e.g., an if-else block in the programming language C always has the same structure. For demonstration purposes, we will use two simpler languages: $L_3 = \{a^n b^n : n > 0\}$, $L_4 = \{\text{sequences of nested parentheses and brackets}\}$. *Context-sensitive grammars* have rules that depend on the position in the sequence—we will use the standard example of $L_5 = \{a^n b^n c^n : n > 0\}$ and $L_6 = \{\text{sequences of paired, but not necessarily nested parentheses and brackets}\}$. We omitted the recursively enumerable grammars similar to Deletang et al. [2022], as they require an infinite tape to simulate, which is impossible.

### 2.2 Out-of-distribution (OOD) generalization

In modern deep learning theory, the test loss distinguishes the performance of models with low training loss by evaluating the model on unseen data sampled from the same distribution (i. e., i.i.d.) as the data it was trained on. When the test loss is (near-)minimal, the model has statistical generalization ability. Therefore several studies focused on establishing bounds on the generalization gap [Vapnik and Chervonenkis, 1971, Dziugaite and Roy, 2017, Pérez-Ortiz et al., 2021]. Along with the question of whether the test loss is sufficiently low, another one arose: does the test loss have a unique minimum? Identifiability is a property of a family of statistical models, concerning the uniqueness of the data generator model recovered from the observed data. In machine learning, identifiability implies the uniqueness of the test loss' minimum, and the model it corresponds to, which is desirable since it enables us to interpret the model and reason about its properties.

Theoretical tools such as statistical generalization or identifiability are mostly concerned about the i.i.d. scenario, i.e., when the training and test data come from the same distribution. However, this is an unrealistic assumption for language models (LMs), especially when pretrained models are used for various downstream tasks. Despite the clear OOD nature of these tasks, OOD generalization of these models is not understood theoretically. Recently, several works addressed a special type of OOD generalization called compositional generalization in vision models [Schott et al., 2021, Wiedemer et al., 2023b,a, Brady et al., 2023, Yang et al., 2023, Lachapelle et al., 2023]; however, such studies are only started emerging for natural language [Ahuja and Mansouri, 2024, Han and Padó, 2024, Ramesh et al., 2024, Lake and Baroni, 2023, Nogueira et al., 2021, Dziri et al., 2023, Saparov et al., 2023]. Deletang et al. [2022] and Ruoss et al. [2023] conducted a similar experimental investigation to ours, the tasks they evaluate on are also derived from formal language recognition and thus grouped according to the Chomsky hierarchy, but they focus on length generalization.

Reizinger et al. [2024] show that despite any explicit inductive bias or regularization, Transformers can exhibit much-better-than-chance extrapolation performance on some synthetic grammars. However, it is unclear whether this behavior is specific to the Transformer and/or the formal language. Furthermore, there are some tasks such as addition and parity that are known to be very hard (or even impossible) to solve by Transformers, at least without tricks [Zhou et al., 2023]. Inspired by these works, our paper investigates the role of architecture in different formal languages.

**Rule extrapolation.** To understand the OOD behavior in AR LMs, we study a particular notion of OOD generalization, which we term *rule extrapolation*. Rule extrapolation is a subclass of compositional generalization, for formal languages are defined by composing multiple rules. When assessing rule extrapolation, the model is pre-trained on formal language data, i.e., the support is the intersection of all language rules. Then, OOD data is presented, where a subset of rules is violated, thus having zero probability over the training distribution. If the completed OOD prompts satisfy the not violated rules, we say the model extrapolates the rules. For example, the $a^n b^n$ language is the intersection of two rules: (R1) the number of $a$'s equals the number of $b$'s and (R2) $a$'s precede $b$'s. The prompt **bbaab** cannot be completed to obey the second rule. In this case, rule extrapolation means that the completed prompt satisfies the first rule (e.g., **bbaab**$a$).

## 2.3 Inductive biases in sequence models

Several deep learning architectures, such as CNNs or GNNs, were designed to capture specific structural data properties. Such inductive biases in sequence models remain to be understood [Reizinger et al., 2024]. McCoy et al. [2020] and Murty et al. [2023] studied whether different architectures on language processing tasks have an inductive bias towards hierarchical structure. [Murty et al., 2023] showed that with sufficient training, the transformer architecture can represent hierarchical sentence structure and use this structure to generalize correctly. Several works establish forms of simplicity bias [Valle-Pérez et al., 2019, Dingle et al., 2018, Mingard et al., 2020]. Goldblum et al. [2023] demonstrate that (even randomly initialized) language models are biased towards low algorithmic complexity. Weiss et al. [2021] developed a formal programming language called RASP to model the inner workings of the Transformer, whereas Zhou et al. [2023] defined a subset, called RASP-L, and proved length generalization in Transformer, emphasizing a simplicity bias in terms of RASP-L code length. Chen et al. [2024] attribute the development of grammatical capabilities to Syntactic Attention Structure (SAS), wherein specific Transformer heads tend to focus on specific syntactic relations. These approaches leverage the tools of theoretical computer science to reason about the success of Transformers, hinting at the role of a structural inductive bias. For example, in-context learning (ICL) performance depends on the ordering of layers in the Transformer [Press et al., 2020], and also the structure of the training data [Chan et al., 2022]. LM inductive biases have also been studied from a mechanistic interpretability perspective. Most notably, Olsson et al. [2022] propose that ICL is due to induction heads (a type of specialised attention heads). Mechanistic interpretability approaches can also identify and disable the computational circuits responsible for bad behaviors [Li et al., 2024] and locate ones that capture factual knowledge [Meng et al., 2023]. These works constitute important progress; though we take a step back to ask: is the good performance attributable to the Transformer? Are (at least some of) these emergent capabilities present in simpler models such as linear models or RNNs?

## 3 Experimental setup

### 3.1 Architectures

To study when rule extrapolation emerges, we compare five architectures: linear models, LSTMs [Hochreiter and Schmidhuber, 1997], Transformers [Vaswani et al., 2023], and State Space Models (SSMs) (focusing on Mamba [Gu and Dao, 2023]), and the recently introduced xLSTM [Beck et al., 2024]. The Transformer [Vaswani et al., 2023] caused a breakthrough in Natural Language Processing (NLP) by introducing the (self-)attention mechanism, allowing it to capture global dependencies efficiently in both directions, unlike the standard LSTM. Adapted from dynamical systems, SSMs have recently entered language modeling, and became increasingly popular, such as this work's focus, Mamba [Gu and Dao, 2023]. In this architecture, the attention mechanism (where every token must "attend" to every other token) is replaced by a single SSM block, allowing the model to selectively focus on relevant information. The on-par performance of the Transformer and the SSM along with the removal of the attention block raises the question of whether the SSM also show rule extrapolation abilities. Recently, Beck et al. [2024] proposed an extension of the LSTM, which includes matrix-valued memory cells, new gating and memory mixing mechanisms, and several computational improvements. Training details, data set sizes, and model parameters are in Appx. B.

### 3.2 Datasets

Our data sets follow the hierarchy of [Chomsky, 1956]. The advantage of the classification is that the categories exhibit fundamental differences. However, this hierarchy is based on computational linguistics concepts. Therefore, there might be no connection between the language's complexity in the Chomsky hierarchy and what a neural network finds difficult to learn. Each language we study obeys two rules, and the OOD prompts violate the corresponding R2, but the prompt can still be completed to satisfy the other. Following (R1) and/or (R2) provide different information: following (R1) means the LM still adheres to a rule even when the other is violated (in the whole sequence), whereas adhering to (R2) on the completion shows that the LM still tries to satisfy that.

The used formal languages and their categorization and rules are included in Tab. 1. We define two rules for each language to keep the results comparable; however, we acknowledge that these can lead to rules of different complexity (cf. the chance levels for $L_3$ and $L_5$ in Tabs. 4 and 6), and also that the rules can potentially be defined in multiple equivalent ways.

**Regular grammars.** Regarding the hierarchy, the two simplest data sets are *regular* languages $L_1 = \{b\alpha : \alpha \text{ contains even number of 'a's}\}$ and $L_2 = \{b^n a^{2m} : n, m > 0\}$. The rules of the language $L_1$ are: (R1) there are even number of $a$s in the sequence; and (R2) the sequence starts with a $b$. For $L_1$, the OOD prompts consist of prompts that violate (R2), i.e. start with an $a$, but all these

prompts can be completed to satisfy (R1). For language $L_2$, the rules are: (R1) there are even number of $a$s in the sequence; and (R2) $b$s precede $a$s. The OOD prompts for $L_2$ violate (R2). They start with a single $a$, then a block of $b$s and possibly a block of $a$s.

**Context-free grammars.** We implemented two *context-free* grammars $L_3$ and $L_4$: $L_3 = \{a^n b^n : n > 0\}$, i.e., (R1) the number of $a$s and $b$s match; and (R2) $a$s precede $b$s. For $L_3$, OOD prompts violate (R2) , i.e., the prompts include $b$ tokens followed by $a$ tokens.

Our fourth formal language is a bracketing (Dyck-) language, i.e., $L_4 = \{$sequences of nested and paired parentheses and brackets$\}$, e.g. "( [ ]( ) )" The rules of the language are: (R1) brackets are nested and paired; and (R2) parentheses are nested and paired. Paired means that every opening bracket/parenthesis has a closing pair; nested means that between an opening and closing bracket/parenthesis, all other tokens must be paired—contrast this with $L_6$. For $L_4$, ID prompts begin with "([" and OOD prompts start with ")["; both are followed by a sequence where the parentheses and the square brackets are matched.

**Context-sensitive grammars.** We implemented two *context-sensitive* grammars $L_5$ and $L_6$. $L_5 = \{a^n b^n c^n : n > 0\}$. Though it seems very similar to $L_3$, its grammar rules make it context-sensitive, i.e., the tokens generated depend on multiple tokens. The grammar rules can be summarized as: (R1) the number of $a$s, $b$s, and $c$s are the same; (R2) $a$s precede $b$s and $b$s precede $c$s; and The OOD prompts are sequences which violate (R2) . All these prompts can still be completed to obey (R1). $L_6$ is a context-sensistve Dyck-language, i.e., $L_6 = \{$sequences of paired, but not necessarily nested parentheses and brackets$\}$, e.g. "( [ )]" The rules of the language are: (R1) brackets are paired; and (R2) parentheses are paired. Akin to $L_4$, for $L_6$ ID prompts begin with "([" and OOD prompts start with ")["; both are followed by a sequence where the parentheses and the square brackets are matched.

### 3.3 Metrics.

We monitor training and test loss. We evaluate the accuracy of both rules (R1/R2) separately and simultaneously both for in-distribution samples, and also for OOD prompts. As OOD prompts are designed that (R2) cannot be satisfied, we evaluate its accuracy in the most lenient way. That is, we either calculate it on the completion or, for the Dyck languages, on the part after the closing parenthesis ")". An example for the $L_3$ OOD prompt **abbb** is as follows: the completion **abbb**$aa$ is considered correct for (R2), but **abbb**$abaa$ is not, as it has an $a$ after a $b$ in the *completion*. Our evaluation is restricted to prompt completions with an EOS token. We also monitor the accuracy of the next token prediction via greedy decoding (i.e., using the token with the largest probability). Our results report the minimum of the test loss to measure whether the models are in the saturation regime [Reizinger et al., 2024]. We select the *largest* values for the rule accuracies. We choose this evaluation as small variations in the test loss could lead to large deviations (as predicted by Liu et al. [2023]). We also report chance level accuracies as a baseline, quantifying how complex a given rule is. Chance level accuracy in each case refers to the performance of a model that always predicts each token (excluding the start-of-sequence (SOS) token) as the next token with equal probability[2]. We report means and standard deviations across 5 seeds. Similar to [Rajamanoharan et al., 2024], we provide a non-representative human baseline based on a small pilot study, where participants have seen three examples for $L_1, L_3$ then were asked to complete five OOD sequences for each (Appx. B.6). We corrected for invalid answers and emphasize that we only aim to provide a sense of how humans measure against neural networks, without reaching any statistical conclusions.

## 4 Results

**Regular grammars.** Perhaps surprisingly, modern architectures perform the worst on regular languages $L_1$ (Tab. 2) and $L_2$ (Tab. 3) : both Mamba and the Transformer are worse in- and out-of-distribution than the LSTM—the xLSTM only matches the LSTM in OOD performance on (R1). Furthermore, the Transformer's accuracies are below chance level even for in-distribution, despite having approximately the same test loss as the LSTM and Mamba. The Linear model seemingly manages to obey perfectly (R2) in-distribution on $L_2$, which happens because this model only predicts EOS on test prompts, and the ID test prompt already satisfies (R2). In the other categories, Linear is at or below chance-level. In our small pilot study, humans performed akin to Mamba on $L_1$ (Tab. 14). Zhou et al. [2023] observed that Transformers struggle with addition or parity calculation, which might explain the Transformer's low performance on regular languages, as both $L_1, L_2$ require calculating the parity of $a$ tokens.

---

[2]The code for calculating chance levels is in `chance_level_accuracies.ipynb`

| Model | Test loss | ID R1 | OOD R1 | OOD R2 completion |
|---|---|---|---|---|
| Chance | N/A | 0.500 | 0.500 | 0.333 |
| Linear | $4.553_{\pm 0.290}$ | $0.500_{\pm 0.000}$ | $0.500_{\pm 0.000}$ | $0.000_{\pm 0.000}$ |
| LSTM | $0.276_{\pm 0.007}$ | $\mathbf{0.926}_{\pm \mathbf{0.110}}$ | $\mathbf{0.862}_{\pm \mathbf{0.143}}$ | $\mathbf{0.947}_{\pm \mathbf{0.107}}$ |
| Mamba | $0.274_{\pm 0.006}$ | $0.634_{\pm 0.130}$ | $0.591_{\pm 0.063}$ | $0.597_{\pm 0.246}$ |
| Transformer | $0.277_{\pm 0.005}$ | $0.393_{\pm 0.402}$ | $0.445_{\pm 0.461}$ | $0.468_{\pm 0.515}$ |
| xLSTM | $0.284_{\pm 0.008}$ | $0.740_{\pm 0.221}$ | $0.679_{\pm 0.183}$ | $0.701_{\pm 0.301}$ |

Table 2: **Test loss and rule-following accuracies for the regular language $L_1 = \{b\alpha\}$**: the LSTM can extrapolate (R1) the best. The column **R2** is left out as it is satisfied by design.

| Model | Test loss | ID R1 | ID R2 | OOD R1 | OOD R2 completion |
|---|---|---|---|---|---|
| Chance | N/A | 0.473 | 0.250 | 0.500 | 0.750 |
| Linear | $1.927_{\pm 2.537}$ | $0.422_{\pm 0.034}$ | $1.000_{\pm 0.000}$ | $0.513_{\pm 0.045}$ | $0.000_{\pm 0.000}$ |
| LSTM | $0.037_{\pm 0.000}$ | $\mathbf{1.000}_{\pm \mathbf{0.000}}$ | $\mathbf{1.000}_{\pm \mathbf{0.000}}$ | $\mathbf{1.000}_{\pm \mathbf{0.000}}$ | $0.000_{\pm 0.000}$ |
| Mamba | $0.038_{\pm 0.000}$ | $0.901_{\pm 0.088}$ | $\mathbf{1.000}_{\pm \mathbf{0.000}}$ | $0.959_{\pm 0.076}$ | $\mathbf{0.073}_{\pm \mathbf{0.120}}$ |
| Transformer | $0.039_{\pm 0.000}$ | $0.158_{\pm 0.357}$ | $0.182_{\pm 0.405}$ | $0.067_{\pm 0.214}$ | $0.000_{\pm 0.000}$ |
| xLSTM | $0.037_{\pm 0.000}$ | $0.833_{\pm 0.408}$ | $0.833_{\pm 0.408}$ | $\mathbf{1.000}_{\pm \mathbf{0.000}}$ | $0.000_{\pm 0.000}$ |

Table 3: **Test loss and rule-following accuracies for the regular language $L_2 = \{b^n a^{2m}\}$**: the LSTM and the xLSTM can extrapolate (R1) the best, closely followed by Mamba

**Context-free grammars.** On the context-free grammars $L_3, L_4$, the conclusion is different. On $L_3$ (Tab. 4), although all four models achieve perfect accuracy on (R2) both in- and out-of-distribution, and all models except the Linear, (near) perfectly obey (R1) in-distribution, the Transformer extrapolates (R1) to the largest extent (66%), followed by the LSTM (38%) and Mamba (30%). The seemingly perfect (R2) ID and OOD extrapolation for the Linear model is, again, due to EOS token generation. On the Dyck language $L_4$ (Tab. 5), the Transformer has the best extrapolation performance, and Mamba is better than the LSTM. On $L_3$, the human participants in our small study had performed better on following (R2) on the completion than extrapolating (R1); however, the Transformer was better than humans in extrapolating both (R1) and (R2).

| Model | Test loss | ID R1 | ID R2 | OOD R1 | OOD R2 completion |
|---|---|---|---|---|---|
| Chance | N/A | 0.105 | 0.356 | 0.154 | 0.445 |
| Linear | $2.553_{\pm 0.159}$ | $0.200_{\pm 0.000}$ | $1.000_{\pm 0.000}$ | $0.275_{\pm 0.000}$ | $1.000_{\pm 0.000}$ |
| LSTM | $0.019_{\pm 0.000}$ | $\mathbf{1.000}_{\pm \mathbf{0.000}}$ | $\mathbf{1.000}_{\pm \mathbf{0.000}}$ | $0.376_{\pm 0.209}$ | $\mathbf{1.000}_{\pm \mathbf{0.000}}$ |
| Mamba | $0.019_{\pm 0.000}$ | $\mathbf{1.000}_{\pm \mathbf{0.000}}$ | $\mathbf{1.000}_{\pm \mathbf{0.000}}$ | $0.296_{\pm 0.043}$ | $\mathbf{1.000}_{\pm \mathbf{0.000}}$ |
| Transformer | $0.022_{\pm 0.002}$ | $\mathbf{1.000}_{\pm \mathbf{0.000}}$ | $\mathbf{1.000}_{\pm \mathbf{0.000}}$ | $\mathbf{0.657}_{\pm \mathbf{0.162}}$ | $\mathbf{1.000}_{\pm \mathbf{0.000}}$ |
| xLSTM | $0.019_{\pm 0.000}$ | $\mathbf{1.000}_{\pm \mathbf{0.000}}$ | $\mathbf{1.000}_{\pm \mathbf{0.000}}$ | $0.438_{\pm 0.252}$ | $\mathbf{1.000}_{\pm \mathbf{0.000}}$ |

Table 4: **Test loss and rule-following accuracies for the context-free language $L_3 = \{a^n b^n\}$**: the Transformer can extrapolate (R1) the best.

**Context-sensitive grammars.** The grammar $L_5$ (Tab. 6) is similar to $L_3$, i.e., the Transformer performs best. Intuitively, the sequences in the form of $\{a^n b^n\}$ and $\{a^n b^n c^n\}$ are rather similar, despite the latter being context-sensitive in Chomsky's hierarchy. Rule extrapolation accuracies for (R1) in $L_5$ are lower than for $L_3$, which can be attributed to the higher complexity of (R1) in the context-sensitive grammar (cf. chance levels in Tabs. 4 and 6). For the context-sensitive Dyck language $L_6$ (Tab. 7), the Transformer and LSTM perform similarly on both OOD (R1) and (R2).

**Results summary.** We conclude that on different grammars, different architectures perform best (Fig. 1). Although the Transformer has a consistently good performance on the investigated context-free and -sensitive grammars, LSTM and Mamba are better choices for the studied regular grammars. We hypothesize that it happens because these languages require calculating parity, in which the Transformer struggles [Zhou et al., 2023]. The xLSTM generally lies somewhere between the LSTM and the Transformer. The Linear model has very limited capabilities for modeling formal grammars as it cannot even minimize the test loss. In our small pilot study on $L_1, L_3$, humans found the tasks difficult: they performed better than chance, though the LSTM performed better on $L_1$, and the Transformer on $L_3$ (Tab. 14)—we emphasize that our human-machine comparison only provides intuition, rather than a rigorous evaluation of human performance, which is left for future work.

| Model | Test loss | ID R1 | ID R2 | OOD R1 | OOD R2 completion |
|---|---|---|---|---|---|
| Chance | N/A | $0.127$ | $0.127$ | $0.127$ | $0.382$ |
| Linear | $6.145_{\pm 0.647}$ | $0.000_{\pm 0.000}$ | $0.000_{\pm 0.000}$ | $0.000_{\pm 0.000}$ | $1.000_{\pm 0.000}$ |
| LSTM | $0.266_{\pm 0.014}$ | $0.961_{\pm 0.075}$ | $0.969_{\pm 0.050}$ | $0.543_{\pm 0.282}$ | $\mathbf{1.000_{\pm 0.000}}$ |
| Mamba | $0.277_{\pm 0.014}$ | $0.697_{\pm 0.152}$ | $0.607_{\pm 0.140}$ | $0.644_{\pm 0.164}$ | $0.886_{\pm 0.129}$ |
| Transformer | $0.273_{\pm 0.018}$ | $\mathbf{0.974_{\pm 0.148}}$ | $\mathbf{0.973_{\pm 0.109}}$ | $\mathbf{0.980_{\pm 0.090}}$ | $\mathbf{1.000_{\pm 0.000}}$ |
| xLSTM | $0.273_{\pm 0.013}$ | $0.706_{\pm 0.116}$ | $0.665_{\pm 0.155}$ | $0.689_{\pm 0.164}$ | $\mathbf{0.991_{\pm 0.018}}$ |

Table 5: **Test loss and rule-following accuracies for the context-free Dyck language $L_4$**: the Transformer can extrapolate (R1) the best.

| Model | Test loss | ID R1 | ID R2 | OOD R1 | OOD R2 completion |
|---|---|---|---|---|---|
| Chance | N/A | $0.022$ | $0.454$ | $0.003$ | $0.593$ |
| Linear | $2.657_{\pm 0.383}$ | $0.000_{\pm 0.000}$ | $0.000_{\pm 0.000}$ | $0.000_{\pm 0.000}$ | $0.000_{\pm 0.000}$ |
| LSTM | $0.017_{\pm 0.001}$ | $\mathbf{1.000_{\pm 0.000}}$ | $\mathbf{1.000_{\pm 0.000}}$ | $0.068_{\pm 0.036}$ | $\mathbf{1.000_{\pm 0.000}}$ |
| Mamba | $0.017_{\pm 0.000}$ | $\mathbf{1.000_{\pm 0.000}}$ | $\mathbf{1.000_{\pm 0.000}}$ | $0.099_{\pm 0.010}$ | $\mathbf{1.000_{\pm 0.000}}$ |
| Transformer | $0.024_{\pm 0.003}$ | $\mathbf{1.000_{\pm 0.000}}$ | $\mathbf{1.000_{\pm 0.000}}$ | $\mathbf{0.187_{\pm 0.085}}$ | $\mathbf{1.000_{\pm 0.000}}$ |
| xLSTM | $0.017_{\pm 0.000}$ | $\mathbf{1.000_{\pm 0.000}}$ | $\mathbf{1.000_{\pm 0.000}}$ | $0.116_{\pm 0.058}$ | $\mathbf{1.000_{\pm 0.000}}$ |

Table 6: **Test loss and rule-following accuracies for the context-sensitive language $L_5 = \{a^n b^n c^n\}$**: the Transformer can extrapolate (R1) the best

| Model | Test loss | ID R1 | ID R2 | OOD R1 | OOD R2 completion |
|---|---|---|---|---|---|
| Chance | N/A | $0.127$ | $0.127$ | $0.127$ | $0.382$ |
| Linear | $4.013_{\pm 0.254}$ | $0.000_{\pm 0.000}$ | $0.000_{\pm 0.000}$ | $0.000_{\pm 0.000}$ | $1.000_{\pm 0.000}$ |
| LSTM | $0.645_{\pm 0.019}$ | $\mathbf{0.981_{\pm 0.042}}$ | $\mathbf{0.956_{\pm 0.061}}$ | $\mathbf{1.000_{\pm 0.000}}$ | $\mathbf{0.894_{\pm 0.165}}$ |
| Mamba | $0.675_{\pm 0.018}$ | $0.745_{\pm 0.070}$ | $0.807_{\pm 0.185}$ | $0.684_{\pm 0.159}$ | $0.810_{\pm 0.212}$ |
| Transformer | $0.640_{\pm 0.016}$ | $\mathbf{1.000_{\pm 0.000}}$ | $\mathbf{1.000_{\pm 0.000}}$ | $\mathbf{0.980_{\pm 0.045}}$ | $\mathbf{0.973_{\pm 0.044}}$ |
| xLSTM | $0.671_{\pm 0.021}$ | $0.791_{\pm 0.179}$ | $0.765_{\pm 0.155}$ | $0.767_{\pm 0.158}$ | $0.715_{\pm 0.121}$ |

Table 7: **Test loss and rule-following accuracies for the context-sensitive Dyck language $L_6$**: the Transformer and the LSTM can extrapolate the best

## 5 Normative theory of OOD prompt completion

The previous sections empirically assessed an example of rational OOD prompt completion: rule extrapolation. In this section, instead of asking what happens, we take a step back to ask what *should* happen: how an ideal model should learn and extrapolate rules. We propose a non-parametric prior and prediction scheme for OOD prompt completion, that can be seen as a generalization of Solomonoff induction [Solomonoff, 2001, Li and Vitányi, 1997] to settings relevant for AR LMs. Although our algorithm, just like Solomonoff induction, is uncomputable, we argue that it formalises a rational approach capable of OOD extrapolation in AR sequence models. Rather than a practical algorithm itself, it should be interpreted as a guide towards building and assessing future practical models. Our conceptual approach is not without precedent: ideas from AIT have recently been popularized as "North Stars" for guiding practical implementations [Theis, 2024, Goldblum et al., 2023], and have been applied in practical algorithms [Grau-Moya et al., 2024].

We first introduce our approach on the high-level, via the following story.

**A story of OOD prompt completion.** Suppose that Bob has a Language Model $p_{\text{data}}$, that autoregressively generates $M$ i.i.d. sequences of length $m$, $\{(x_{1,j}, x_{2,j}, \ldots x_{m,j})\}_{j=1}^{M} := (x_{1j}^{m})_{i=1}^{M}$. Since the sequences are generated autoregressively, we may call $(x_{1j}^{m-1})_{j=1}^{M}$ the *ID prompt*s, and each $m^{th}$ element $(x_{m,j})_{j=1}^{M}$ its *ID completion*s. Suppose that Charlie, Bob's enemy, generates a $n-$length sequence from the same LM, and intervenes (in the causal sense) on it, so that the resulting sequence $(x_1, x_2, \ldots x_{n-1}) := x_1^{n-1}$ has zero probability under the LM. We call this the *OOD prompt*. Despite $p_{\text{data}}(x_1^{n-1}) = 0$, the LM still defines the conditional probability of completing the OOD prompt $x_1^{n-1}$. Charlie then asks an observer, Alice, to predict how Bob's LM will complete the OOD prompt $x_1^{n-1}$, i.e., what $x_n$ will be. Fig. 2 shows the probabilistic assumptions of Alice: the completions are generated independently, according to the same procedure (i.e., using the same LM). We use the conditional independence assumption $x_n \perp (x_1^m)_{j=1}^{M} \mid x_1^{n-1}$ in eq. (4) below.

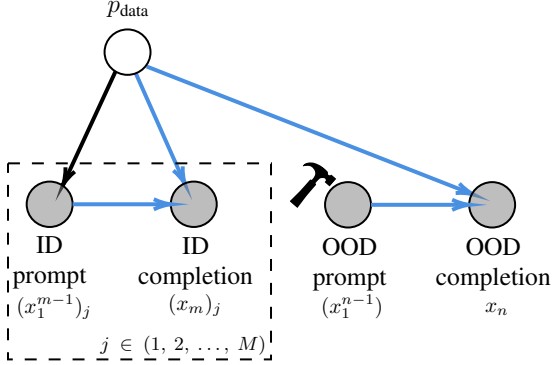

Figure 2: Graphical model representing our approach for OOD prompt completion. Although Bob's LM $p_{\text{data}}$ assigns zero probability to the OOD prompt, it defines a conditional probability distribution for its completions. Our probabilistic model assumes that Bob's LM completes the ID and OOD prompt independently, according to the same procedure (e.g. the same LM architecture and parameters are used for generating the completions). This is the same as assuming that the Markov factors marked in blue are the same, i.e. $p(\text{completion}|\text{prompt}, p_{\text{data}}) = p(\text{completion}|\text{OOD prompt}, p_{\text{data}})$, and the conditional independence OOD completion $\perp$ ID prompt $|$ OOD prompt.

In the rest of this section, we construct an algorithmic prior that formalizes these assumptions, and argue why it is a promising approach to study OOD compositional generalization theoretically.

### 5.1  The Solomonoff prior

The Solomonoff prior assigns a prior probability to individual data points based on some algorithmic notion of how difficult it is to generate that data point. It embodies Occam's razor and Epicure's principle, as simple data points have a larger probability, and every possible explanation is included in the prior (see also Appx. C.2). For simplicity, we define the Solomonoff prior for discrete sample spaces, though similar arguments hold for the continuous case. To encourage readability, we define technical terms in Appx. C.2, and highlight them in blue here. Let us fix a monotone universal Turing machine (UTM). Solomonoff's universal prior [Solomonoff, 2001] is defined over arbitrary-length sequences $x_1^N := (x_1, x_2, \dots, x_N)$ as

$$p_S(x_1^N) = \sum_i \alpha(p_i) p_i(x_1^N), \tag{1}$$

where we sum over all discrete lower semicomputable semimeasures $p_i(x_1^N)$ implementable on the UTM [Li and Vitányi, 1997]. We will refer to the $p_i(x_1^N)$ as mixture components or *explanations* of the data. The prior on weights $\alpha(p_i)$ is an arbitrary semimeasure, i.e., $\forall i : \alpha(p_i) > 0$ and $\sum_i \alpha(p_i) \leq 1$. Frequently, $\alpha(p_i)$ is chosen as $2^{-K(p_i)}$, the prefix Kolmogorov complexity of $p_i$ in the UTM (see Defn. C.5 in Appx. C.2).

**Predictive form.** The above formulation of the Solomonoff prior has the predictive form [Hutter, 2005, Chapter 3.2.3], where $\alpha(p_i \mid x_1^{N-1})$ is updated via Bayesian inference:

$$p_S(x_N \mid x_1^{N-1}) = \sum_i \alpha(p_i \mid x_1^{N-1}) p_i(x_N \mid x_1^{N-1}), \text{ where } \alpha(p_i \mid x_1^{N-1}) = \frac{\alpha(p_i) p_i(x_1^{N-1})}{p_S(x_1^{N-1})} \tag{2}$$

**Convergence of predictions.** Suppose that the true distribution of $(x_1, x_2, \dots, x_N)$ is $\mu$. The Solomonoff prior (with any valid sequence of weights) satisfies [Hutter, 2005].

$$p_S(x_N \mid x_1^{N-1}) \xrightarrow{N \to \infty} \mu(x_N \mid x_1^{N-1}) \text{ with } \mu-\text{probability 1}. \tag{3}$$

### 5.2  A predictive model for OOD prompt completion

Our goal is to define a similar prior, and predictive scheme that fits our scenario of AR next-token prediction, and where we can express the notion of completing an out-of-distribution prompt $x_1^{n-1}$, even when our prior assigns zero probability to the prompt.

The Solomonoff prior assigns nonzero prior mass to every possible prompt, i.e. there exist no OOD problems for the Solomonoff prior, as each possible test distribution is included in the prior as a mixture component $p_i$. However, by definition, the Solomonoff prior can only take in a single sequence $x_1^n$. This means that it can only model pre-training and (OOD) testing together, since the pre-training and testing data need to be concatenated into the same sequence [Hutter, 2011]. Intuitively, it is more natural to separate those processes. To achieve this, we propose an adapted version of the Solomonoff prior, modifying it two ways, and justifying our approach below:

(i) We condition the prediction on a pre-training dataset $\mathcal{D}$ of $M$ independent and identically distributed (i.i.d.) sequences of finite length $m$, i.e. $\mathcal{D} = \{x_{1j}^m\}_{j=1}^M$. $\mathcal{D}$ is sampled from the distribution $p_{\text{data}}^M(\mathcal{D}) = \prod_{j=1}^M \prod_{k=2}^m p_{\text{data}}(x_{k,j} \mid x_{1,j}^{k-1})$. For simplicity, we assume that each pre-training datapoint has equal length $m$.

(ii) Instead of modelling semimeasures as joints over sequences $\{(x_1^N)\}_{N \in \mathbb{N}}$, we model semimeasures as lists of conditionals, just as how AR LMs model probability distributions over $\{(x_k \mid x_1^{k-1})\}_{k=2}^N$, enumerating them with index $i = 1, 2, \ldots$, denoting each semimeasure as $p_{i|}$ to emphasize the lists of conditionals representation. That is, $p_{i|}(x_k \mid x_1^{k-1})$ and $p_{i|}(\mathcal{D})$ mean $p_i(x_k \mid x_1^{k-1})$ and $p_i(\mathcal{D}) = \prod_{j=1}^M \prod_{k=2}^m p_i(x_{k,j} \mid x_{1,j}^{k-1})$, respectively. Note that the pre-training distribution $p_{\text{data}}$ also belongs to the set of $p_{i|}$. We define a mixture over all lists of discrete lower semicomputable semimeasures $p_{i|}$ implementable on the UTM See Appx. C.1 for details.

The motivation for modelling $x_1^N$ as a list of conditionals is because the mapping from lists of conditional factorizations to joint semimeasures consistent with them is a many-to-one mapping, because zero-probability sequences have multiple factorizations (see Appx. C.1 for justification and more details on this notation). If the prompt $x_1^{n-1}$ comes from a distribution different from $\mathcal{D} \sim p_{\text{data}}^M$, that assigns zero probability mass to $x_1^{n-1}$, the probability $p_{\text{data}}(x_n \mid x_1^{n-1})$ is left undefined if only the joint probability $p_{\text{data}}(x_1^n)$ is specified. This is not a problem in the Solomonoff prior, as it assigns nonzero probability mass to every (computable) sequence. But once we introduce the conditioning on $\mathcal{D}$, this step becomes necessary. The above two modifications generalize the predictive form of the Solomonoff prior as follows (we color-code the equation denoting modification (i) in red and modification (ii) in green):

$$p_R(x_n \mid x_1^{n-1}, \mathcal{D}) := \sum_i \alpha(p_{i|} \mid \mathcal{D}) p_{i|}(x_n \mid x_1^{n-1}), \text{ with } \alpha(p_{i|} \mid \mathcal{D}) = \frac{\alpha(p_{i|}) p_{i|}(\mathcal{D})}{p_{\text{data}}(\mathcal{D})}. \quad (4)$$

**Interpreting $p_R$.** Starting from a prior weight $\alpha$ over all possible explanations $p_{i|} = \{p_i(x_k \mid x_1^{k-1})\}_{k=2}^n$, the posterior probability of $p_{i|}$ given $\mathcal{D}$ is computed (eq. (4), right). The $n^{th}$ step prediction by $p_i$, conditioned on a possibly OOD test prompt, is then weighted by this posterior. It is important that the prediction $p_{i|}(x_n \mid x_1^{n-1})$ is *not* conditioned on the pre-training data $\mathcal{D}$, and the posterior $\alpha(p_{i|} \mid \mathcal{D})$ is *not* conditioned on the test prompt $x_1^{n-1}$. This, as stated above, separates pre-training from testing, enabling us to define the completion of OOD test prompts. When $\mathcal{D}$ equals $x_1^{n-1}$, $p_R$ reduces to $p_S$, and thus the posterior prediction converges according to eq. (3).

**Choice of the weight prior $\alpha(p_{i|})$.** For OOD test prompts, there are multiple explanations $p_{i|}$ consistent with $\mathcal{D}$. Therefore, the behaviour of $p_R$, even when $|\mathcal{D}|$ tends to infinity, depends on the prior weight $\alpha(p_{i|})$. This differs from the Solomonoff prior, which converges to the true posterior regardless of the weights (eq. (3)) [Hutter, 2005]. Thus, $\alpha$ must be chosen to allow the extrapolation of simple explanations consistent with the data. We define $\alpha(p_{i|}) := 2^{-K(p_{i|})}$, penalising exponentially the length of the shortest program (implemented on the fixed UTM) $K(p_{i|})$ that can approximate $p_{i|}$ (each conditional probability) for every prompt $x_1^n$. This encodes Occam's razor into the prior, and is consistent with the optimal weights of the Solomonoff prior [Hutter, 2005].

## 5.3 Towards explaining training dynamics and rule extrapolation

Here, we argue informally that our normative algorithm provides a notion of a rational pre-training process, and thus helps explain the training dynamics of practical LMs, and is also capable of rule extrapolation. We support our arguments by showing the role of simplicity bias (towards low Kolmogorov complexity) in the dynamics of learning the $a^n b^n$ language with Transformers.

**Explaining training dynamics.** We analize the dynamics of learning rule extrapolation. We report results on the Transformer (training dynamics of Mamba and the LSTM are in Appx. A), trained on the $a^n b^n$ language—where high rule extrapolation ability is achieved. Fig. 3 shows that first, the model learns the sequences obeying (R2), then it learns the language (R1) ∩ (R2) as its subset.

We argue that the order in which rules are learnt is governed by the relative simplicity of the rules, quantified by Kolmogorov complexity. Given a formal language with rules (R1) and (R2), let $p_1$, $p_2$ and $p_{1,2}$ be distributions defined by LMs that generate sequences that satisfy (R1), (R2) and (R1) ∩ (R2), respectively. If, e.g., $K(p_2) \ll K(p_{1,2})$, our normative algorithm will first learn (R2), and then learn the (R1) ∩ (R2) as its subset. In the $a^n b^n$ language, (R2) ($a$'s before $b$'s), is, on average, simpler to generate than (R1) (#a=#b) and (R1) ∩ (R2). Therefore, we expect our normative algorithm to first learn (R2), and then learn (R1) ∩ (R2) as its subset. Remarkably, our Transformer employs the same

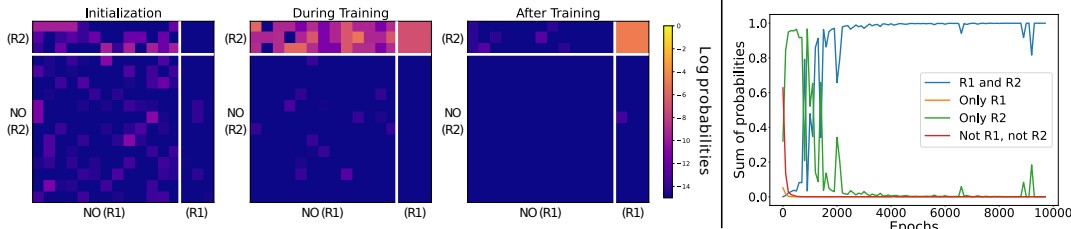

Figure 3: **Training dynamics of rule learning for a Transformer trained on the $a^n b^n$ language:** we color-code the log probability of all sequences of length $8$ consisting of $a$'s and $b$'s and ending with EOS at initialization (**left** *left*), during (**left** *middle*) and after training (**left** *right*). The sequences are separated according to which rule they obey. While at initialization, the probabilities are distributed roughly evenly, during training the model starts to assign higher probabilities to sequences satisfying (R2). After training the most likely sequences are the ones in (R1) $\cap$ (R2), the others are negligible. The same trend can be seen on the **right**, where the normalized sum of the probabilities of the four categories (satisfying (R1) and (R2), only (R1), only (R2) and neither) is plotted during training.

strategy ( Fig. 3), verifying the presence of simplicity bias. This result is matches past observations that Transformers are biased towards low Kolmogorov complexity [Goldblum et al., 2023].

**Towards explaining rule extrapolation.** Our normative algorithm has been designed to complete OOD prompt based on the simplest explanations consistent with the pre-training data. On the high level, this approach is consistent with rule extrapolation. We conjecture that approximating our normative algorithm similarly to the approach of Grau-Moya et al. [2024], will result in models with superior rule extrapolation properties. We leave this promising direction to future work.

## 6 Discussion

**Conclusion.** We argue that focusing on rule extrapolation and formal languages gives us sound (theoretical) tools to analyze and better understand out-of-distribution behaviour in language models, such as the role of different architectures. Our empirical findings emphasize that no single universal architecture exists for autoregressive sequence modeling. Though Transformers fare very well in most scenarios we investigated, they struggled on regular languages. Therefore, we argue that the architecture's inductive bias should be considered when selecting models since the architecture that performs the best depends on the nature of the task. Furthermore, we analyse the training process enabling rule extrapolation, we find that the model first identifies the whole set obeying one of the rules, then it learns the language (intersection of all rules) as its subset. Beyond advancing our empirical understanding, we also proposed a normative theory of OOD prompt completion. Our normative algorithm predicts the next token based on simple explanations consistent with the data, and allows us to explain and contextualise some of our empirical observations.

**Impact.** Rule extrapolation is a special case of compositional generalization in language models. While other OOD generalisation types were examined previously, this is the first work studying rule extrapolation. This novel concept has the potential to impact LLM research both on conceptual and practical levels. General compositional generalization notions examine whether from learning multiple concepts/rules separately, the model can understand the composition of the concepts/intersection of the rules. However, in rule extrapolation, we measure the reverse direction: from the composition/intersection, can the model identify the concepts/rules separately? Importantly, this direction is less straightforward. Rule extrapolation allows for easy study of compositional generalisation ability on a variety of datasets, such as formal or programming languages. Therefore rule extrapolation has the potential to become an established benchmark task for evaluating current and future LM architectures.

**Limitations.** We defined and empirically evaluated rule extrapolation in simple formal languages, where analysis is tractable and demonstrates that models can "go beyond" their training data. We acknowledge that our data sets are far from natural language where rule extrapolation may be difficult to demonstrate. Studying formal languages may still have practical relevance, e. g. for programming languages or formal mathematics. Even though we considered different hyperparameter setups presented in the appendix, we have not performed exhaustive ablations over the hyperparameters or analysis of architectures. Furthermore, model variants, like different attention or positional encoding, may impact our findings.

## Acknowledgements

The authors would like to thank Gergely Flamich for several inspiring discussions on Solomonoff induction, Bence Nyéki for insights on practical aspects of natural language processing and Gail Weiss for her insights on PCFGs. This work was supported by a Turing AI World-Leading Researcher Fellowship G111021. Patrik Reizinger acknowledges his membership in the European Laboratory for Learning and Intelligent Systems (ELLIS) PhD program and thanks the International Max Planck Research School for Intelligent Systems (IMPRS-IS) for its support. This work was supported by the German Federal Ministry of Education and Research (BMBF): Tübingen AI Center, FKZ: 01IS18039A. Wieland Brendel acknowledges financial support via an Emmy Noether Grant funded by the German Research Foundation (DFG) under grant no. BR 6382/1-1 and via the Open Philantropy Foundation funded by the Good Ventures Foundation. Wieland Brendel is a member of the Machine Learning Cluster of Excellence, EXC number 2064/1 – Project number 390727645. This research utilized compute resources at the Tübingen Machine Learning Cloud, DFG FKZ INST 37/1057-1 FUGG.

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

# A    Further experimental results on training dynamics

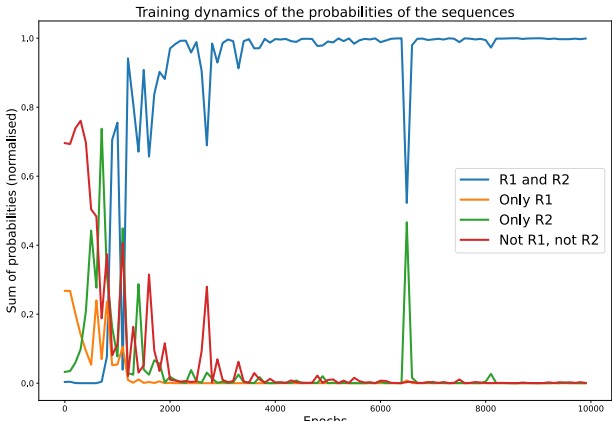

Figure 4: **Training dynamics of the LSTM** Training an LSTM on the $a^n b^n$ language, the normalized probability of all sequences, grouped into the four categories (satisfying (R1) and (R2), only (R1), only (R2) and neither) of length $8$ consisting of $a$'s and $b$'s and ending with EOS is plotted during training. The sequences are separated according to which rule they obey. At initialization, sequences obeying any of the rules have low probability. During training, the model first starts assigning higher probabilities to sequences satisfying (R2), but soon after, sequences in (R1) ∩ (R2) dominate. After training the most likely sequences are the ones in (R1) ∩ (R2), the others are negligible.

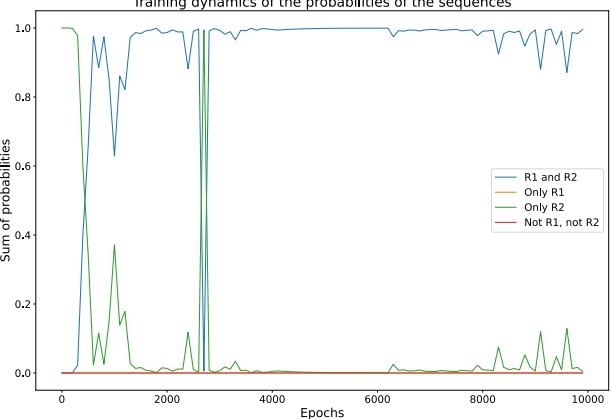

Figure 5: **Training dynamics of Mamba** Training a Mamba architecture on the $a^n b^n$ language, the normalized probability of all sequences, grouped into the four categories (satisfying (R1) and (R2), only (R1), only (R2) and neither) of length $8$ consisting of $a$'s and $b$'s and ending with EOS is plotted during training. The sequences are separated according to which rule they obey. Intriguingly, at initialization, sequences obeying (R2) are assigned largest probability. During training, the model learns (R1) ∩ (R2) consistently after 3000 epochs. After training the most likely sequences are the ones in (R1) ∩ (R2), the others are negligible.

# B Experimental details

## B.1 Reproducibility and codebase.

We use PyTorch [Paszke et al., 2019], PyTorch Lightning [Falcon, 2019], and HuggingFace Transformers [Wolf et al., 2020]. Our training pipeline builds on [Reizinger et al.] and we use the PyTorch implementation of Mamba from [LeGuet] and the code released by the authors for the xLSTM [Beck et al., 2024]. Our code and experimental logs are publicly available at `https://github.com/meszarosanna/rule_extrapolation`.

## B.2 Formal grammars

**Training data.** We generate data from the formal languages $L_1$, $L_2$, $L_3$, $L_4$ and $L_5$ described in § 3.2 up to length 256—excluding the SOS and EOS tokens, i.e., those two tokens add two to the maximal length. The SOS (0), EOS (1), and padding (2) tokens are always represented by these numbers. When the grammar consists of letters, their representations are $a$ (3), $b$ (4) and $c$ (5), and when the language is the nested brackets and parentheses the tokens are the $'('$ (3), $')'$ (4), $'['$ (5) and $']'$ (6).

We used different data set sizes for the different languages. This is explained by the highly different size of all possible sequences that obey all rules of any language. For the languages, $L_1$, $L_2$ the training set consists of 15000 samples (as these languages have rules satisfied by many sequences), and for $L_4$, 512 samples. For $L_3$ and $L_5$, the corresponding data sets include all unique sequences up to length 256, which is 128 for $L_3$ and 85 for $L_5$, respectively.

**Test prompts.** We define our test prompts as all possible sequences of length 8 (prepended with SOS) for $L_1$ and $L_3$, and all possible sequences of length 5 (prepended with SOS) for $L_5 = \{a^n b^n c^n : n > 0\}$—we chose different lengths to have a comparable number of test samples, i.e., $2^8$ and $5^3$, respectively. We split these sets into in-distribution and OOD test prompts, based on whether they can be completed to obey the rules of the specific grammar.

For $L_2$, first, we generate in-distribution test prompts of length 8—these can be completed according to the grammar rules by definition. From these, we create the OOD prompts by adding a single $a$ to the beginning of the sequences. For $L_4$, we sample length-6 sequences obeying both rules, then the ID prompts are prepended with $'($ $['$ and the OOD prompts with $')$ $['$. Then the prompts are prepended with SOS.

## B.3 Model and training parameters

We observed that the Linear model constantly predicts PAD tokens, unless we ignore those by setting the `ignore_index=PAD` in `torch.nn.CrossEntropyLoss()`. However, for comparison, when reporting the losses, we report the loss where we do not set the `ignore_index` parameter.

Table 8: General parameters

| PARAMETER | VALUES |
|---|---|
| TRAINING DATA MAXIMUM LENGTH | 256 |
| PROMPT PREDICTION CUTOFF LENGTH | 300 |
| BATCH SIZE | 128 |
| OPTIMIZER | ADAMW |
| LEARNING RATE SCHEDULER | INVERSE SQUARE ROOT |
| BATCH SIZE | 128 |
| LEARNING RATE | 2e−3 |
| NUMBER OF EPOCHS | 50,000 |

Table 9: Linear model parameters

| PARAMETER | VALUE |
|---|---|
| MODEL | LINEAR |
| DIMENSION OF THE MODEL | 256 |
| BIAS | TRUE |

Table 10: LSTM parameters

| PARAMETER | VALUE |
|---|---|
| MODEL | STANDARD LSTM |
| NUMBER OF LAYERS | 5 |
| EMBEDDING DIMENSION | 16 |
| HIDDEN DIMENSION | 64 |
| DROPOUT PROBABILITY | 0.4 |

Table 11: Transformer parameters

| PARAMETER | VALUE |
|---|---|
| MODEL | TRANSFORMER DECODER |
| NUMBER OF LAYERS | 7 |
| MODEL DIMENSION | 10 |
| NUMBER OF ATTENTION HEADS | 5 |
| FEEDFORWARD DIMENSION | 1024 |
| DROPOUT PROBABILITY | 0.1 |
| LAYER NORM $\epsilon$ | 6e$-$3 |
| ACTIVATION | ReLU |

Table 12: Mamba parameters

| PARAMETER | VALUE |
|---|---|
| MODEL | MAMBA |
| NUMBER OF LAYERS | 10 |
| MODEL DIMENSION | 32 |
| DIM OF CONV LAYER | 8 |
| DIM OF STATE SPACE | 16 |

Table 13: xLSTM parameters[3]

| PARAMETER | VALUE |
|---|---|
| MODEL | xLSTM |
| NUMBER OF BLOCKS | 6 |
| EMBEDDING DIMENSIONS | 64 |
| mLSTM CONV1D KERNEL SIZE | 4 |
| mLSTM $qkv$ PROJECTION BLOCK SIZE | 4 |
| mLSTM NUMBER OF HEADS | 4 |
| sLSTM POSITION | 1 |
| sLSTM NUMBER OF HEADS | 4 |
| sLSTM CONV1D KERNEL SIZE | 4 |
| sLSTM BIAS INITIALIZATION | BLOCK-DEPENDENT POWER LAW |
| sLSTM FEEDFORWARD PROJECTION FACTOR | 1.3 |
| sLSTM FEEDFORWARD ACTIVATION | GeLU |

## B.4 Training dynamics plot generation details

Figure 3 was plotted on the $a^n b^n$ language with the Transformer on seed 63656. The **left** *left* was plotted at the Lightning module's "self.global_step=0", **left** *middle* at "self.current_epoch = 600" and **left** *middle* at nearly end of the training at "self.current_epoch=9700". On the **right**,

---

[3]Adopted from `https://github.com/NX-AI/xlstm?tab=readme-ov-file#xlstm-language-model`

the sum of the probabilities was computed at every epoch divisible by 100. Similarly, Figure 4 was plotted on $a^n b^n$ with LSTM with seed 8556, and Figure 5 on $a^n b^n$ with Mamba with seed 91686.

## B.5 Additional experimental results

**Greedy decoding vs sampling.** Our initial results use greedy decoding, but we conducted experiments to evaluate the sampling method for next token prediction. As shown in Fig. 6, we conclude that while the the Transformer is the best choice with greedy decoding, except for regular languages where the LSTM performs better (Fig. 6a); the LSTM appears to excel when using sampling (Fig. 6b). These results also open up new interesting future directions, e.g., investigating the influence of different temperature values in the softmax.

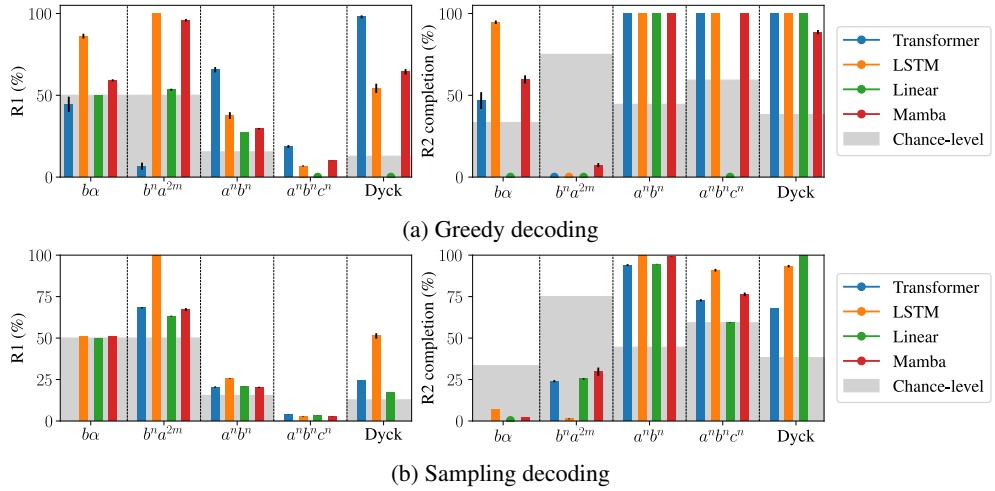

(a) Greedy decoding

(b) Sampling decoding

Figure 6: **Rule extrapolation summary for all models but the xLSTM and languages $L_1 - L_5$ (Tab. 1) with *greedy* (Fig. 6a) and *sampling* (Fig. 6b) next-token decoding**

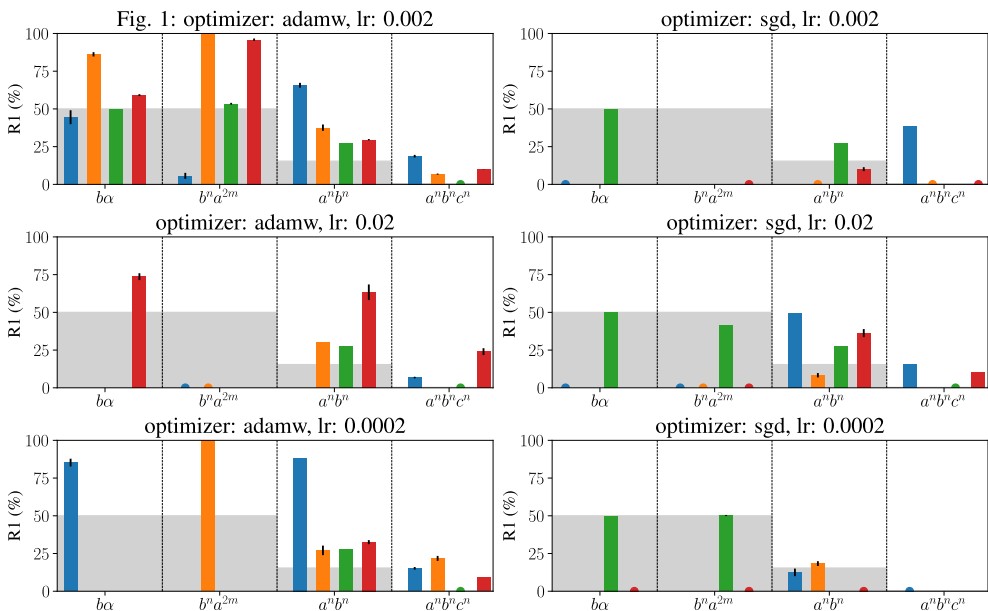

Figure 7: **Rule extrapolation performance for different optimizers and learning rates for all models except the xLSTM and languages $L_1 - L_5$ (Tab. 1):** top row left is the same as Fig. 6a

**Hyperparameter sensitivity.** We tested multiple hyperparameters, including three learning rates and two optimization algorithms, and plotted the results in Fig. 7. Though our hyperparameter search

is not exhaustive, we can state that when considering the best settings for each architecture, the LSTM consistently performs best on regular languages, while the Transformer excels on everything else.

**Model size ablation.** We tested varying size settings (different numbers of layers and heads) for the Transformer architecture to determine whether increasing size can improve performance on regular languages. As shown in Fig. 8, increasing the Transformer model size does not meaningfully improve performance on regular languages; the best values remain those originally used (`num_layers = 7`, `num_heads = 5`). For non-regular languages, the Transformer already outperformed the other architectures.

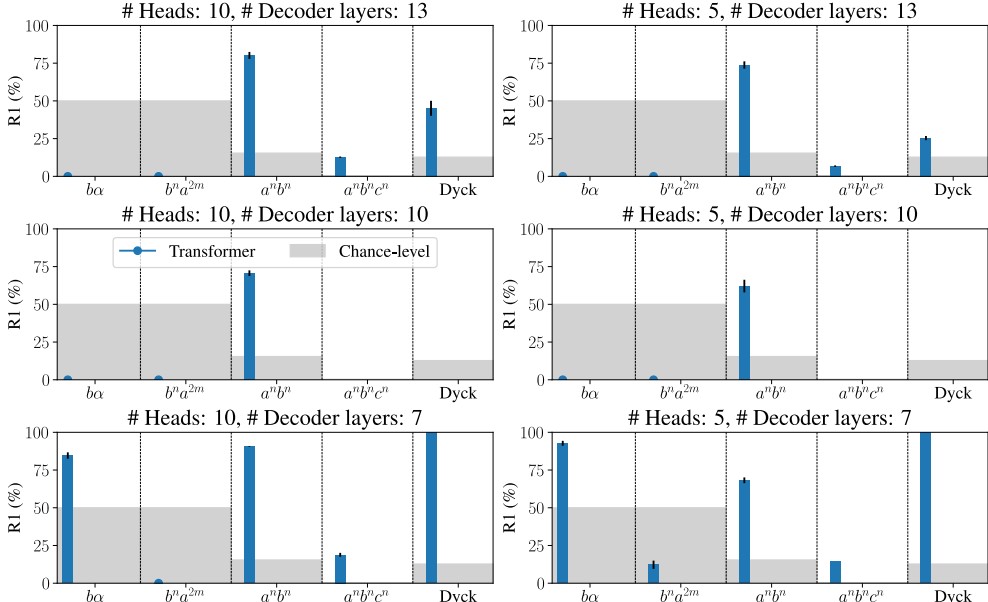

Figure 8: **Rule extrapolation performance for different number of attention heads and decoder layers in the Transformer for languages** $L_1 - L_5$ **(Tab. 1)**

### B.6 Human pilot study details

We conducted a small pilot study with humans using an online questionnaire (the study size was 14). We did not collect any personal information, only task-relevant responses.

**Instructions.** The participants received the following instruction:
*This questionnaire asks you to perform a task of completing sequences based on examples for 2 cases. Each case follows the same layout:*

- *first, we show some example sequences*
- *then on, we start sequences and ask you to finish them as you see fit*

*Then, on the three following pages of the questionnaire, they were presented the following:*
*We generate sequences according to some patterns. You see below examples, which are considered completed (whitespace is only for visibility reasons):*
[Examples came here in the questionnaire; detailed below]
*Now you will see 5 incomplete messages. What you see are the first characters of sequences of unknown length. Your task is to finish them.*
*When writing down your answer:*

- *DO NOT include the beginning of the sequence already provided, only your completion of it.*
- *The length of your answer is up to you, choose what you see fit.*
- *If you think the sequence is already completed, leave the space for the completion empty*

**Case 1: regular grammar** $L_1$**.** The examples for the context-free grammar $L_1$ were:

- $baba$
- $baa$

- *babbaaa*

The prompts the participants needed to complete were:

- *aba*
- *abba*
- *abaaba*
- *abab*
- *aaba*

**Case 2: context-free grammar $L_3$.** The examples for the context-free grammar $L_3$ were:

- *aabb*
- *aaabbb*
- *aaaaaabbbbb*

The prompts the participants needed to complete were:

- *baa*
- *abaa*
- *bba*
- *baab*

**Results.** We preprocessed the questionnaire results to remove invalid responses (e.g., those with invalid characters, where we assumed that we did not explain the task well to the study subjects). We report the OOD (R1) and (R2) accuracies, the latter only on the completion in Tab. 14.

| Language | OOD R1 | OOD R2 completion |
|---|---|---|
| $L_1 = \{b\alpha\}$ | 0.654 | 0.605 |
| $L_3 = \{a^n b^n\}$ | 0.415 | 0.623 |

Table 14: **Human pilot study OOD accuracies:** humans in our study performed better than chance, though they could not beat the LSTM on $L_1$ and the Transformer on $L_3$

### B.7 Computational requirements

Our models and data sets are small scale and were designed to fit into an NVIDIA GeForce RTX 2080 Ti with 11GB VRAM, this guided our parameter choices (Appx. B.3). As we used SLURM and Condor managed clusters, our experiments were, due to GPU availability, in some cases, allocated on NVIDIA A100 GPUs. Although in the paper we report statistics over 5 seeds, in some cases, we ran more experiments during the lifetime of the project. For transparency, we report overall numbers, given in GPU hours for each synthetic grammar (Tab. 1). The runtimes differed based on model architecture, data set size, and the stochasticity of the training (i.e., the use of early stopping)

- $L_1$ : 1,455 GPU hours for 107 runs
- $L_2$ : 707 GPU hours for 59 runs
- $L_3$ : 301 GPU hours for 334 runs
- $L_4$ : 269 GPU hours for 208 runs
- $L_5$ : 264.5 GPU hours for 65 runs,

which amounts to approximately 3,000 GPU hours and also includes the GPU hours required for creating the figures (Fig. 3).

To estimate the energy consumption, we take the maximum power consumption of an NVIDIA A100 (PCIe version), which is 250W[4]. This amounts to approximately 750kWh, which is equivalent to the emission of 0.313 metric ton $CO_2$, i.e., approximately 1290 kilometres driven by an average gasoline-powered passenger vehicle[5].

---

[4] `https://www.nvidia.com/content/dam/en-zz/Solutions/Data-Center/a100/pdf/nvidia-a100-datasheet.pdf`

[5] `https://www.epa.gov/energy/greenhouse-gas-equivalencies-calculator`

## C   Details on the normative theory of OOD extrapolation

### C.1   The set of joints and conditional factorizations

In this section, we denote sets of probability distributions as $D$ and use subscripts $J$ and $C$ to refer to joint and conditional distributions, respectively. Consider the set of joint probabilities on $N-$length sequences, where the $p_i$ are drawn from some set $S$. For example, $S = \{p_i :$ is computable w.r.t. a UTM$\}$.

$$D_{J,N} = \{p_i(x_1, x_2, ..., x_N), \ p_i \in S\} \tag{5}$$

and the set of conditional factorizations consistent i.e., such conditionals that the joint equals the product of the conditionals, with them

$$D_{C,N} = \{\{p_{i|}(x_k \mid x_1^{k-1})\}_{k=1}^N, \text{ consistent with elements of } S\}. \tag{6}$$

We claim that $D_{J,N} \subset D_{C,N}$. To see that $D_{J,N} \subseteq D_{C,N}$, note that the list $\{p_{i|}(x_k \mid x_1^{k-1})\}_{k=1}^N$ uniquely determines $p_i(x_1, x_2, ..., x_N)$ as the product of its elements. To see that the sets are not equal, consider the following example.

**Example.**   Let $X_1$ and $X_2$ be two binary random variables. Let us define a probability mass function (pmf) $p$ such that $p(X_1 = 0) = 0$ and $p(X_1 = 0, X_2 = 0) = p(X_1 = 0, X_2 = 1) = 0$. Now consider two sets of conditional pmfs $q_1$ and $q_2$, satisfying

$$q_1(X_2 = x \mid X_1 = 1) = q_2(X_2 = x \mid X_1 = 1) = p(X_2 = x \mid X_1 = 1) = p(X_2 = x),$$

$$q_1(X_1 = 1) = q_2(X_1 = 1) = 1,$$

but

$$q_1(X_2 = 0 \mid X_1 = 0) = 1 \text{ and } q_2(X_2 = 0 \mid X_1 = 0) = 0.$$

Due to the first two equations, both $q_1$ and $q_2$ are consistent with $p$, but they can differ on the zero-probability prompt $X_1 = 0$.

Hence the set corresponding to $D_{C,N}$ is larger, and the extra elements correspond to the zero-probability sequences under each $p_i$. These are precisely the prompts on which we assess rule extrapolation.

**The lists of conditionals notation.**   In § 5, we distinguish between the joint probability representation $p_k := \{p_k(x_1^N)\}$ and the lists of conditionals representation $p_{i|}$. Let $\phi$ denote the mapping from lists of conditionals to the joint probabilities. Consider the set of pre-images of $p_k$ under $\phi$, i.e., $\phi^{-1}(p_k)$, which has cardinality $|\phi^{-1}(p_k)|$. If this set has multiple elements, we can enumerate them as $\{p_{k|,j}\}_{j=1}^{|\phi^{-1}(p_k)|}$, with $p_{k|,j} := \{p_{k,j}(x_k \mid x_1^{k-1})\}_{k=1}^N$, where $p_{k,j}$ is the $j^{th}$ element of $\phi^{-1}(p_k)$. In our predictive $p_R$, we list the $p_{i|}$, where the index $i$ is understood to loop over all pre-images: $\{p_{i|}\}_i \equiv \{\{p_{k|,j}\}_{j=1}^{|\phi^{-1}(p_k)|}\}_k$, where the enumerations over $(k, j)$ are combined into an enumeration over $i$ in a dovetail fashion. Index $k$ loops over the joint probability distributions, and $j$ loops through each of their pre-images. Note that this is a different enumeration than the one in the Solomonoff prior $p_S$, where only the joint probabilities are enumerated (here with index $k$).

### C.2   Solomonoff Induction

This section has been adapted from Li and Vitányi [1997], Hutter [2005] and Hutter [2011].

**Epicure's principle**   states that if more than one theory is consistent with the observations, one should keep all the theories. The Solomonoff prior follows this principle in including all (lower semicomputable) semimeasures in the prior.

**Occam's razor**   states to keep the simplest theory consistent with the observations. The Solomonoff prior follows this in assigning larger probabilities to algorithmically more complex strings.

**Definition C.1** (**Prefix code**). A prefix code $P$ is a set of binary strings such that no element is proper prefix of another. It satisfies Kraft's inequality $\sum_{p \in P} 2^{-l(p)} \le 1$.

**Turing machines.**    A Turing machine can be thought of as an idealised form of a computer. Informally, it consists of tapes, read/write heads, a table of rules and an internal state. There are multiple technical variants of Turing machines. Here, we define prefix Turing machines.[6]

**Definition C.2 (Prefix Turing Machine).** A prefix Turing machine $T$ is a Turing machine with one unidirectional (i.e. the head can only move from left to right) input tape, one unidirectional output tape, and some bidirectional work tapes. Input tapes are read only, output tapes are write only. All tapes are are binary (no blank), work tapes are initially filled with zeros.

We say that $T$ *halts* on input $p$ with output $x$, and write $T(p) = x$ if $p$ is to the left of the input head and $x$ is to the left of the output head after $T$ halts. The set of $p$ on which $T$ halts forms a prefix code. We call such codes $p$ *self-delimiting* programs. The Turing machine may take another input $y$ on its input tape. Since $T$ is a prefix Turing machine, $y$ needs to be prefix encoded, denoted as $y`$, and then concatenated to the program $p$. In this case, we say $T(y`p) = x$.

The table of rules of a Turing machine $T$ can be encoded as a binary string, which we denote by $\langle T \rangle$. Hence the set of Turing machines $\{T_1, T_2, \dots\}$ can be enumerated (computably). We will use this property when we sum over Turing machines.

**Universal Turing Machines.**    There are so-called universal Turing machines, which can "simulate" all Turing machines. We define a particular one which simulates a prefix Turing machine $T(q)$ if fed with input $\langle T \rangle q$, i.e. $U(\langle T \rangle q) = T(q) \ \forall T, q$. If $p$ is not of the form $\langle T \rangle q$, $U(p)$ does not output anything. We call this particular $U$ the *reference universal Turing machine*.

**Semimeasures.**    Let $\mathcal{X}^*$ be the set of finite strings and $\mathcal{X}^\infty$ be the set of infinite sequences over some alphabet $\mathcal{X}$ of size $|\mathcal{X}|$. Recall our sequence notation from § 5: for a string $(x_1, x_2, \dots, x_n) \in \mathcal{X}^*$ of length $n$ we write use the shorthand $x_1^n$ with $x_i \in \mathcal{X} \quad \forall i \in \{1, 2, \dots, n\}$.

**Definition C.3 (Semimeasure).** Let $\epsilon$ denote the empty string. A function $\mu : \mathcal{X}^* \to \mathbb{R}$ is a semimeasure if for all $x \in \mathcal{X}^*$, $\mu(\varepsilon) \leq 1$, and $\mu(x) \geq \sum_{b \in \mathcal{X}} \mu(xb)$, where $xb$ denotes the concatenation of $x$ and $b$, also an element of $\mathcal{X}^*$. If equalities hold, $\mu$ is called a probability measure.

**Remark C.1.** $p_S$ and $p_R$ (§ 5) are semimeasures, because $\sum_{x_1^n} p_S(x_1, x_2, ..., x_n) < 1$. The fact that the integral is less than 1 is due to the halting problem of UTMs [Turing, 1936], which means that there are some programs in the sum that never stop running.

**Definition C.4 (Lower semicomputability).** A function $f : \mathbb{N} \to \mathbb{R}$ is lower semicomputable iff there exists a computable function $\phi(x, k) : \mathbb{Q} \times \mathbb{N} \to \mathbb{Q}$, such that

- $\lim_{k \to \infty} \phi(x, k) = f(x)$
- $\forall k \in \mathbb{N} : \phi(x, k + 1) \geq \phi(x, k)$.

i.e, if it can be approximated from below to arbitrary precision.

**Kolmogorov complexity.**    Kolmogorov complexity measures the complexity of an object as the length of the shortest program that generates the object. There is also a conditional version, based on the length of programs that input some other objects.

**Definition C.5 ((Conditional) prefix Kolmogorov complexity).** The (conditional) prefix Kolmogorov complexity of a string $x$ is the length $l$ of the shortest halting program $p$ for which $U$ outputs $x$ (given $y$):

$$K(x) := \min_p \{l(p) : U(p) = x \text{ halts}\}. \tag{7}$$

$$K(x|y) := \min_p \{l(p) : U(y`p) = x \text{ halts}\}. \tag{8}$$

The Kolmogorov complexity of a semimeasure, $p_i(x_1^n)$, is understood to be the length of the shortest self-delimiting program on $U$, computing $p_i(x_1^n)$ given $x_1^n$, for every $x_1^n$.

---

[6]Some works introduce the Solomonoff prior using monotone Turing machines [Hutter, 2011, Grau-Moya et al., 2024], but for our purposes, using prefix Turing machines is equivalent [Li and Vitányi, 1997].

## D Acronyms

**AR LM** autoregressive language model

**CS** context-sensitive

**EOS** end-of-sequence

**i.i.d.** independent and identically distributed
**ICL** in-context learning

**LM** language model

**NLP** Natural Language Processing

**OOD** out-of-distribution

**RASP** Restricted-Access Sequence Processing
Language

**SOS** start-of-sequence
**SSM** State Space Model

