# OpenReview forum: "Rule Extrapolation in Language Modeling: A Study of Compositional Generalization on OOD Prompts"
_NeurIPS.cc/2024/Conference — NeurIPS 2024 spotlight_

### Official Review · Reviewer_AxjF · 2024-07-01

**Soundness:** 4
**Presentation:** 4
**Contribution:** 4
**Rating:** 6
**Confidence:** 3

**Summary:**

This paper studies rule extrapolation, one OOD behavior, of autoregressive LLMs on different models to understand the effect of model's architecture on this specific ability. The paper also introduces a normative theory for OOD prompt completion, which well explains the empirical observation about the training dynamics enabling the rule extrapolation.

**Strengths:**

The paper studies an interesting and important question for language models: how different models do extrapolation and how well they are. It is well-structured and clearly presented. The authors conduct extensive and well-designed simulations, yielding valuable insights into the rule extrapolation capabilities of various models. Additionally, the paper introduces a normative theory to elucidate the training dynamics associated with rule extrapolation observed in practice. This is a very good starting point for the community to investigate the OOD problem for the language models.

**Weaknesses:**

The paper only investigates four models with fixed sizes. It is commonly acknowledged that the model capacity rely on the model size, which is not adequately addressed in the paper. While the study provides a general impression of the models' capabilities on different tasks, it remains unclear whether the observed differences are due to variations in model size, specific structural characteristics of each model, or a combination of both.

**Questions:**

I have several concerns:
1. Does the model size significantly impact performance? For instance, if the size of the Transformer model is increased, is there a potential for a substantial improvement in its performance on regular grammars?
2. Contemporary language models typically use sampling rather than greedy decoding to generate sequences. If the sampling technique is altered, would the results remain consistent?

**Limitations:**

See weakness and questions.

---

> ### Author Rebuttal · Authors · 2024-08-07
>
> We would like to thank the reviewer for their positive evaluation of our work, in particular highlighting its relevance, good structure, clarity and deeming our experiments extensive and well-designed.
>
> We agree with the reviewer that beyond the present experiments, there are various other options and ablations to evaluate. We have carried out additional experiments on:
> - Transformer model size
> - Sampling instead of greedy decoding
> - Hyperparameter ablation
> - A new context-sensitive language
> - The xLSTM architecture
>
> Please find the results of your suggested experiments under ‘Transformer size ablation’ and ‘Sampling’ below. The results of the rest of the experiments are in [our general response](https://openreview.net/forum?id=Li2rpRZWjy&noteId=1xbJMvUYnC).
>
> ## **Transformer size ablation**:
>  As suggested, we tested varying size settings (different numbers of layers and heads) for the Transformer architecture to determine whether increasing size can improve performance on regular languages. As shown in Figure 7(b) (cf. the uploaded pdf), increasing the model size does not meaningfully improve performance on regular languages; the best values remain those originally used (`num_decoder_layers = 7, num_heads = 5`). For non-regular languages, the Transformer already outperformed the other architectures.
>
> ## **Sampling**:
>  Our initial results use greedy decoding, but we conducted experiments to evaluate the sampling method for next token prediction. As shown in Figure 6(a), we conclude that while the Transformer is the best choice with greedy decoding (except for regular languages where LSTM performs better), LSTM appears to excel when using sampling. These results also open up new interesting future directions, e.g., investigating the influence of different temperature values in the softmax.
>
> We hope that our additional experiments have addressed the reviewer’s concerns.

---

> > ### Comment · Reviewer_AxjF · 2024-08-09
> >
> > Thanks for your reply! I am satisfied with the answers and I don't have any additional concerns or questions.

---

> > > ### Author Response · Authors · 2024-08-10
> > >
> > > We are delighted that our answers addressed your concerns. We also appreciate your positive ratings of "excellent" for soundness, presentation and contribution. If possible, we would be grateful if you could consider raising the overall rating. If there are any further improvements we could make to achieve a higher score, we would be more than happy to address them.

---

### Official Review · Reviewer_3gdK · 2024-07-04

**Soundness:** 3
**Presentation:** 3
**Contribution:** 3
**Rating:** 8
**Confidence:** 3

**Summary:**

# Problem:

We lack systematic understanding about the Out-Of-Distribution (OOD) behaviours of autoregressive language models (LMs), such as in-context learning with natural languages prompts, despite successful deployment of LMs in such OOD situations.

Natural languages (NLs) prompts, i.e. real-world data, are acknowledged as too complex to systematically study OOD behaviours.

# Contributions:

Formal languages have obvious practical relevance for programming languages and formal mathematics, despite their dissimilarities with NLs.

In contrast to NLs, formal languages enable studying in a systematic fashion rule extrapolation (as a special case of compositional generalization (systematicity)/OOD behaviours) and therefore provides a systematic framework to analyze and better understand LMs. Thus, the paper defines and empirically evaluate rule extrapolation for simple formal languages, with linear, recurrent (LSTM), transformers and state space models.

The paper also investigates whether the emergent capabilities found in Transformer-based LMs are also present in simpler models?

Transformers models are found to surprisingly struggle with regular languages while they outperform other models with other categories, i.e. context-free and context-sensitive languages.

LSTM and SSM models are found to indeed have some emergent capabilities for OOD behaviours, albeit to a lesser extent than Transformers, and the LSTM struggles less with regular languages.

Finally, the paper proposes a non-parametric prior and prediction scheme for OOD prompt completion using Solomonoff induction, and discusses it in comparison to recent Algorithmic Information Theory frameworks, towards ‘building and assessing future practical models’. Their proposed prior is found to match the training dynamics of the Transformers on rule extrapolation for a context-free language ($L_3=\{a^n b^n\}$).

**Strengths:**

# Strengths:

## Originality:

SO1: The paper is as original as it gets, as far as I am concerned, and I appreciate how it tries to still build bridges with previous work, mainly as it discusses how their normative approach relates to previous Algorithmic Information Theory methods.

## Quality:

SQ1: I appreciate section 2.2.’s discussion between Reizinger et al.[2024] and Zhou et al., [2023], and would encourage the authors to expand further on the impact on this work, possibly in the Discussion section.

## Clarity:

SC1: I appreciate the clarity externalisation approach used in Section 5.1.

## Significance:

SS1: Section 4 - Regular grammars, the paper proposes insights about the Transformer’s surprisingly low performance by relating it to previous work on parity ( but please see WS1 below).

**Weaknesses:**

# Weaknesses:

## Originality:

Nothing to report, I find this paper as original as it gets.

## Quality:

WQ1: section 2.1 does not describe recursively enumerable languages nor explain why they are not considered in this paper. Adding those considerations could be one way to paint a more complete bigger picture about the current work and therefore enhance its quality.

## Clarity:

WC1: section 2.2. starts directly with ‘statistical generalization’ and ‘identifiability’ without having defined them, which might possibly hinders the readability of the paper for readers that are unfamiliar with this (relatively-)recent literature.

WC2: missing ‘that’ or ‘which’ on ln94, possibly --> ‘formal languages _that_ are’ …

WC3: typos on ln208: ‘largest extent(64%)’ --> ‘..(66%)’ ; on ln209 : ‘model again due’ --> ‘model _is_ again due’

WC4: Figure 2 left : it is unclear what is represented given the range of the legend (from 0 to -15), please clarify?

Similarly, in Figure 2 right, I do not understand why is it a _sum_ of probabilities and not just the probability or likelihood, please clarify?

## Significance:

WS1: Following SS1 above, I think the paper could have a greater impact by possibly reproducing the parity experiment with the current architectures and propose correlation measures between the scores on OOD (R1/R2) and the parity accuracy, for instance.

WS2: Providing results for the recently proposed xLSTM would increase the impact of the paper. Thus I would like to encourage the authors to consider adding it.

**Questions:**

# Questions:

Please see above, but mainly:

- WC4

# General advice:

I would like to propose the authors to present section 4.1 after section 5 (theory before experimental validation) for increased impact. Indeed, when reading the paper the first time, I was surprised to find section 4.1 and could not understand what did it brought to the discussion. After having read section 5 it became clear that section 4.1 is an experimental validation to the theoretical concerns of section 5, thus my recommendation to change the ordering.

**Limitations:**

## Limitations:

The paper acknowledges limitations in terms of external validity with respect to different architectures and different attention or encoding mechanism.

# POST REBUTTAL UPDATE :

Most of my concerns have been addressed through the rebuttal in a satisfiable way, thus I am increasing my overall rating to 8, as well as contribution and presentation ratings to 'good'.

---

> ### Author Rebuttal · Authors · 2024-08-07
>
> We warmly thank the reviewer for their positive evaluation of our work, in particular highlighting its originality, clarity and interesting nature. Please find our replies to your comments and suggestions below.
>
> ## WQ1 (section 2.1 does not describe recursively enumerable languages)
>
> We added a discussion of recursivey enumerable languages. We omitted these similar to Deletang et al, 2022, as they require an infinite tape to simulate, which is impossible.
>
> ## SQ1 (Expand discussion about further impact)
>
> Thank you for your feedback, we have expanded on this in our discussion, which we summarise here.
> While other OOD generalisation types were examined in the literature (Ruoss et al, 2022; Deletang et al. 2022; Ahuja and Mansouri, 2024), this is the first work studying/evaluating rule extrapolation. This novel concept has the potential to impact LLM research both on conceptual and practical levels:
> - General compositional generalization notions examine whether from learning multiple concepts/rules separately, the model can understand the composition of the concepts/intersection of the rules. However, in rule extrapolation we measure the reverse direction: from the composition/intersection, can the model identify the concepts/rules separately. Importantly, this notion of compositionality is less straightforward than the generally considered direction.
> - Natural languages are compositional, thus, we expect this property in LLMs if they ought to model these languages well. Therefore, studying rule extrapolation can help us better understand LLMs’ inner workings.
> - Rule extrapolation allows for studying compositional generalisation ability easily on a variety of datasets, such as formal or programming languages. Therefore rule extrapolation has the potential to become an established benchmark task for evaluating current and future LM architectures.
>
> ## WC1 (New paragraph in the background)
>
> We have included a new paragraph in the background section about statistical generalisation and identifiability to improve the readability of the paper.
>
>
> ## WC2-3 (typos)
> Thank you for pointing out the typos, we have corrected them.
>
> ## WC4 (Figure 2)
> On the left, we plotted the log probability, resulting in the rage 0 - (-15). We updated the plot with a label and the caption to clarify this issue. On the right, we intended to show that R2 is learnt first and the language is identified as its subset, which is why we *summed* the probabilities of all sequences obeying only R1, only R2, both R1 and R2, and not R1 nor R2.
>
> ## WS2 (xLSTM)
> We have carried out the suggested experiments with the xLSTM architecture.
> Due to the limitations of our computational resources, not all languages could be tested with a significant number of seeds, and there was no time for extensive hyperparameter ablations. We aim to update our results in the discussion period. Our results are in Figure 6 (b) in the uploaded document. We see that xLSTM indeed outperforms LSTM, but cannot reach the Transformer on the non-regular languages.
>
>
> ### Hyperparameters for xLSTM:
> We used the following hyperparameters for xLSTM. Most of these were suggested by the xLSTM github repository, except for num_blocks and xlstm_embedding_dim, which were lowered to match the size of the other models we trained. This model has 185K parameters.
>  ```model: xlstm
>   mlstm_block:
>     mlstm:
>       conv1d_kernel_size: 4
>       qkv_proj_blocksize: 4
>       num_heads: 4
>   slstm_block:
>     slstm:
>       backend: cuda
>       num_heads: 4
>       conv1d_kernel_size: 4
>       bias_init: powerlaw_blockdependent
>     feedforward:
>       proj_factor: 1.3
>       act_fn: gelu
>   num_blocks: 5
>   xlstm_embedding_dim: 64
>   slstm_at: [ 1 ]
> ```
>
> ## WS1 (Parity Experiment)
> Thank you for this suggestion. We thought about how to best incorporate parity into our framework, and we carried out the following experiment.
>
> **Parity extrapolation.** In the anbn language, Rule 1 (#a=#b) is a subset of the rule “even number of tokens in the sequence”. Thus, in order to extrapolate R1 on OOD prompts where R2 is broken, it’s necessary for the model to also understand and extrapolate parity (since #a=#b means the sequence has even length). To understand the relationship between parity and R1 extrapolation, we tested whether failure on R1 extrapolation also means failure in parity extrapolation.
>
>
> Our results **(see the below table)** show that models learnt to extrapolate parity, despite their imperfect R1 extrapolation accuracies. This supports our intuition  that parity is an easier task, and R1 extrapolation requires other concepts as well (such as equality).
>
>
> | Model                 | Test loss                 | ID R1                     | ID parity                 | OOD R1                    | OOD parity                |
> |--------------------------------|------------------------------------|------------------------------------|------------------------------------|------------------------------------|------------------------------------|
> | Linear | $2.796\scriptscriptstyle\pm 0.171$ | $0.200\scriptscriptstyle\pm 0.000$ | $1.000\scriptscriptstyle\pm 0.000$ | $0.275\scriptscriptstyle\pm 0.000$ | $1.000\scriptscriptstyle\pm 0.000$ |
> | LSTM           | $0.019\scriptscriptstyle\pm 0.000$ | $1.000\scriptscriptstyle\pm 0.000$ | $1.000\scriptscriptstyle\pm 0.000$ | $0.351\scriptscriptstyle\pm 0.056$ | $1.000\scriptscriptstyle\pm 0.000$ |
> | Transformer   | $0.022\scriptscriptstyle\pm 0.002$ | $1.000\scriptscriptstyle\pm 0.000$ | $1.000\scriptscriptstyle\pm 0.000$ | $0.628\scriptscriptstyle\pm 0.103$ | $1.000\scriptscriptstyle\pm 0.000$ |
>
>
> ## General advice
> Thank you for pointing out this unclarity in our narrative. We have changed the structure as suggested: section 5 now precedes section 4.1.
>
> We would like to thank the reviewer again for their constructive questions and suggestions, which we hope we successfully addressed.

---

> > ### Comment · Reviewer_3gdK · 2024-08-14
> > **Reply**
> >
> > I thank the authors for their thorough rebuttal and their careful address of my concerns and recommendations.
> > I am very satisfied with it, and thus increase my overall rating to 8, and contribution and presentation ratings to 'good'.

---

### Official Review · Reviewer_57oB · 2024-07-11

**Soundness:** 3
**Presentation:** 2
**Contribution:** 2
**Rating:** 6
**Confidence:** 2

**Summary:**

This paper studies the compositional generalization of auto-regressive large language models with respect to rule extrapolation in formal languages. Both linear and recurrent architectures, including transformers, are compared on the task of inferring the rules that define regular grammars, context-free grammars, and context-sensitive grammars. The paper also presents the theoretical contribution of a normative theory of rule extrapolation, grounded on the idea of the Solomonoff prior, taken from algorithmic information theory.

**Strengths:**

+ Understanding the compositional generalization capability of large language models is a challenging important task
+ Interesting theoretical framework insipired by the Solomonoff prior and information theory
+ Experimental evaluation with different baselines and languages/grammars

**Weaknesses:**

- Few theoretical insights to justify the experimental results (see comments below)
- One single architecture tested for each model, and one single language for each category

**Questions:**

* The paper makes the conjecture that LSTMs perform better than LLMs on regular languages, because such grammars require comuting parity, which is a notoriously difficul task for LLMs. Did the experimental evaluation consider also different architectures and/or
hyper-parameters for the transformer, to exclude that such worse performance depends on the choice of the model?

* Similarly, the experimental evaluation is conducted on a single language for each category, and for a single architecture for each model: I wonder whether different architectural choices were tried for different models.

* At the end of page 4, in the definition of "Context-sensitive grammar" there is R2 twice, while the first one should be R1.

**Limitations:**

See questions above.

---

> ### Author Rebuttal · Authors · 2024-08-07
>
> We thank the reviewer for their positive evaluation of our chosen topic, proposed theory and experiments. We address your questions below:
>
> >One single architecture tested for each model, and one single language for each category
>
> Our architectures have been selected through initial manual hyperparameter optimisation. However, we have added further new experiments: size, hyperparameter, sampling technique ablations, a new architecture (xLSTM) and a new context-sensitive grammar (non-nested Dyck). Originally, we had 2 languages in each category except in the context-sensitive, but now there are 2 languages in that category, too. Please find the details in the joint response to reviewers, and on the uploaded document. Altogether (including all seeds), we have evaluated 1170 models.
>
> >Did the experimental evaluation consider also different architectures and/or hyper-parameters for the transformer, to exclude that such worse performance depends on the choice of the model?
>
> - From the additional experiments, it can be seen that increasing the **model size** does not meaningfully improve performance on regular languages; the best values remain those originally used (num_layers = 7, num_heads = 5) (Figure 7 (b)).
> - Regarding **hyperparameters** such as the optimisation algorithm and the learning rate, when considering the best settings for each architecture, LSTM consistently performs better than the Transformer on regular languages (Figure 7 (a)).
> - When using **sampling** instead of greedy decoding for next token prediction, LSTM is consistently better than the Transformer in OOD R1 extrapolation (Figure 6 (a)).
>
> >Similarly, the experimental evaluation is conducted on a single language for each category, and for a single architecture for each model: I wonder whether different architectural choices were tried for different models.
>
> Besides the size, optimisation algorithm, learning rate, and sampling technique ablations, we added a **new context-sensitive language** to ensure there is more than one grammar in each category. The new language is the non-nested Dyck language, where brackets and parentheses do not need to be nested; for example, the sequence ([)] is grammatical here. For the results, please see the joint response. We conclude that this language fits our narrative, specifically that the Transformer architecture is usually the best choice for non-regular languages. Figure X shows (cf. the uploaded pdf) the results: the Transformer and LSTM perform similarly on both OOD R1 and R2 completion. We note that although the Linear model appears to be the best, it is not representative since it predicts only the EOS token, resulting in an empty sequence that obeys both rules.
>
> >At the end of page 4, in the definition of "Context-sensitive grammar" there is R2 twice, while the first one should be R1.
>
> Thank you for pointing out the typo; we have corrected it.
>
> We would like to thank the reviewer for their valuable feedback, we hope that the extra experiments and reframed insights address the reviewer’s concerns.

---

> > ### Comment · Reviewer_57oB · 2024-08-12
> > **Rebuttal**
> >
> > I thank the authors for the effort put in the rebuttal and above all for the additional experiments, which strengthen the message of the paper. I have increased my score.

---

### Official Review · Reviewer_Nwv7 · 2024-07-13

**Soundness:** 4
**Presentation:** 3
**Contribution:** 3
**Rating:** 6
**Confidence:** 4

**Summary:**

The article examines systematic generalization in neural networks using artificial grammar learning tasks. Distinctive to this work, the authors operationalize systematic generalization through studying how models extrapolate learned rules to ungrammatical (and thus OOD) seqeunces. Their examination considers a range of models as well as grammars of varying complexity. The article ends with a discussion of algorithmic information theorry and how it relates to OOD generalization.

**Strengths:**

The article has a number of strengths:
- Distinctive operationalization of compositional generalization
- Solid technical methodology
- Evaluated a number of different model architectures and grammars
- The writing and presentation are clear

**Weaknesses:**

The article also has weaknesses:

- In my view the section on "Normative theory of OOD prompt completion" didn't contribute much to the article. It's right that this can serve as a normative account of how models should generalize OOD. However the relationship to the current article seemed thin. Instead, the authors could have established normative baselines for their tasks using probabilistic grammar induction and actually run a model to demonstrate a type of normative generalization.
EDIT : The author's response helps to address this question.

- This is a fine, technically sound article that doesn't strike me as particularly high impact.

**Questions:**

- This statement is just asserted but wasn't obvious to me and could use justification:
"In the a^n b^nn language, R2 (a-s before b-s), is, on average, simpler to generate than R1 (#a=#b) and R1 ∩ R2.
EDIT : The author's response helps to address this question, and should be included in an updated paper.

typos
- "an Bob's enemy" pg. 7

**Limitations:**

This section was fine

---

> ### Author Rebuttal · Authors · 2024-08-07
>
> We thank the reviewer for their positive evaluation of our experiments and presentation. Please find below our replies to your concerns.
>
> ## RE normative theory
> We thank the reviewer for their feedback on our normative theory in Section 5. We agree that our theory relates loosely to the rule extrapolation phenomenon discussed primarily in this article. However, we think that even general (normative) theories of OOD prompt completion are limited in the literature, let alone specialised versions that the reviewer is suggesting. As a first step, we wanted to offer a general perspective on how an idealised model should complete OOD prompts, as we felt that general intuition was missing. Therefore, rather than fully explaining the special case of rule extrapolation, our work offers deeper intuition to the nature of OOD prompt completion as choosing the simplest hypotheses consistent with the training data. Our theory builds on Solomonoff induction, the foundation of Bayesian inference, and offers a new, novel way of relating OOD extrapolation into this general framework.
>
> Apart from providing foundational understanding, we also relate our normative theory to the dynamics of rule learning in our experiments. We argue that the order in which the rules are learned is governed by the simplicity of the rules. This is in line with existing observations on the simplicity bias of language models (towards low Kolmogorov complexity [Goldblum et al., 2023], and further motivates our approach of building on the general theory of Solomonoff induction. In order to strengthen this connection in our narrative, we swapped Section 4.1 (training dynamics experiment) and Section 5 (normative theory), which makes it explicit that the training dynamics experiment is a verification of the high-level ideas in our normative theory section.
>
>
> ## RE simplicity of rules
>
> This high-level statement refers to the Kolmogorov complexities of generating R1 (#a=#b) and R2 (a-s before b-s). The shortest program generating instances following R2 is likely shorter than the shortest program generating instances from R1, since the R2 program can default to outputting only b-s once it generates a b, while the R1 program needs to keep track of the number of a-s and b-s it generates. Furthermore, R1 in itself is a regular language which is accepted by a pushdown automaton, while R2 defines a context-free language which is recognisable by a simpler automata, a finite-state machine.
>
>
> ## RE impact of paper
> Thank you for your feedback. We realise that we have not fully clarified the significance of our work in our original manuscript. We have expanded on this in our discussion, which we summarise here.
>
> While other OOD generalisation types were examined in the literature (Ruoss et al, 2022; Deletang et al. 2022; Ahuja and Mansouri, 2024), this is the first work studying/evaluating rule extrapolation. This novel concept has the potential to impact LLM research both on conceptual and practical levels:
>  - General compositional generalization notions examine whether from learning multiple concepts/rules separately, the model can understand the composition of the concepts/intersection of the rules. However, in rule extrapolation we measure the reverse direction: from the composition/intersection, can the model identify the concepts/rules separately. Importantly, this notion of compositionality is less straightforward than the generally considered direction.
> - Natural languages are compositional, thus, we expect this property in LLMs if they ought to model these languages well. Therefore, studying rule extrapolation can help us better understand LLMs’ inner workings.
> - Rule extrapolation allows for studying compositional generalisation ability easily on a variety of datasets, such as formal or programming languages. Therefore rule extrapolation has the potential to become an established benchmark task for evaluating current and future LM architectures.
>
> In order to further strengthen the significance of our contribution, we added multiple new experiments (xLSTM, size, hyperparameter, sampling technique ablations, and a new context-sensitive language). Please find the details in [our joint response](https://openreview.net/forum?id=Li2rpRZWjy&noteId=1xbJMvUYnC), and in the uploaded document.
>
> ## References
> M. Goldblum, M. Finzi, K. Rowan, and A. G. Wilson. (2023). The no free lunch theorem, Kolmogorov complexity, and the role of inductive biases in machine learning,  URL https://arxiv.org/pdf/2304.05366
>
> K. Ahuja and A. Mansouri. (Feb. 2024) On Provable Length and Compositional Generalization, URL http://arxiv.org/abs/2402.04875
>
> Deletang, G., Ruoss, A., Grau-Moya, J., Genewein, T., Wenliang, L. K., Catt, E., Cundy, C., Hutter, M., Legg, S., Veness, J., & Ortega, P. A. (2022, September 29). Neural Networks and the Chomsky Hierarchy. The Eleventh International Conference on Learning Representations. https://openreview.net/forum?id=WbxHAzkeQcn
>
> Ruoss, A., Delétang, G., Genewein, T., Grau-Moya, J., Csordás, R., Bennani, M., Legg, S., & Veness, J. (2023). Randomized Positional Encodings Boost Length Generalization of Transformers (arXiv:2305.16843). arXiv. https://doi.org/10.48550/arXiv.2305.16843

---

> > ### Comment · Reviewer_Nwv7 · 2024-08-07
> >
> > Thank you for the detailed reply and additional experiments, which I found helpful. I raised my score.

---

### Author Rebuttal · Authors · 2024-08-07

We would like to thank the reviewers for their constructive feedback and suggestions. We were happy to see that **the reviewers found our topic important, our methodology solid, and our presentation clear**. We supply separate replies to all reviewers, but summarize the common points in our joint response.

The main suggestion of the reviewers was to add more experiments. We would like to summarise these here in the joint response. We implemented all the suggested experiments, including a new language, a new architecture, and an extensive hyperparameter search covering model size, optimization algorithm, learning rate, a parity check and next token prediction method. Overall, these experiments confirm our results and narrative. However, due to time and computational constraints, the results are based on a limited number of seeds, and some settings are missing. We will continue these experiments and update the results during the discussion period. If our paper is accepted, we will include all the necessary experiments in the camera-ready version. Currently, altogether (including all seeds), we have evaluated 1170 models. Please find our current conclusions below.


### New context-sensitive language:
 We added a new context-sensitive language to ensure there is more than one grammar in each category. The new language is the non-nested Dyck language, where brackets and parentheses do not need to be nested; for example, the sequence ([)] is grammatical here. We conclude that this language fits our narrative, specifically that the Transformer architecture is usually the best choice for non-regular languages. **The table below** shows the results: the Transformer and LSTM perform similarly on both OOD R1 and R2 completion. We note that although the Linear model appears to be the best, it is not representative since it predicts only the EOS token, resulting in an empty sequence that obeys both rules.

| Model| Test loss | ID R1 | ID R2| OOD R1| OOD R2  completion |
|--|--|----|-|-|---|
| Linear | $4.013\pm 0.254$ | $0.000\pm 0.000$ | $0.000\pm 0.000$ | $0.000\pm 0.000$ | $1.000\pm 0.000$ |
| LSTM           | $0.645\pm 0.019$ | $0.981\pm 0.042$ | $0.956\pm 0.061$ | $1.000\pm 0.000$ | $0.894\pm 0.165$ |
| Mamba          | $0.675\pm 0.018$ | $0.745\pm 0.070$ | $0.807\pm 0.185$ | $0.684\pm 0.159$ | $0.810\pm 0.212$ |
| Transformer   | $0.640\pm 0.016$ | $1.000\pm 0.000$ | $1.000\pm 0.000$ | $0.980\pm 0.045$ | $0.973\pm 0.044$ |





### xLSTM:
As suggested by one of the reviewers, we implemented the latest extension of the LSTM architecture, the xLSTM model. Due to time constraints, we did not conduct a hyperparameter search, however, using a model with approximately the same number of parameters as our other architectures performs in alignment with our narrative. Furthermore, we see that xLSTM indeed outperforms LSTM, but cannot reach the Transformer on the non-regular languages. The results are plotted in Figure 6 (b)  (cf. the uploaded pdf), and we will include the outcomes of additional runs with different hyperparameters in our paper.

### Sampling the next token:
Our initial results use greedy decoding, but we conducted experiments to evaluate the sampling method for next token prediction. As shown in Figure 6 (a), we conclude that while the Transformer is the best choice with greedy decoding (except for regular languages where LSTM performs better), LSTM appears to excel when using sampling. These results also open up new interesting future directions, e.g., investigating the influence of different temperature values in the softmax.

### Hyperparameters:
We tested multiple hyperparameters, including three learning rates and two optimization algorithms, and plotted the results in Figure 7 (a)  (cf. the uploaded pdf).  Due to time constraints, we could not produce results for all languages, but we will include them in the paper as soon as possible; we plan to do so during the discussion period. Despite this, our claims remain valid: when considering the best settings for each architecture, LSTM consistently performs best on regular languages, while the Transformer excels on everything else.

### Size ablation:
As suggested, we tested varying size settings (different numbers of layers and heads) for the Transformer architecture to determine whether increasing size can improve performance on regular languages. As shown in Figure 7 (b)  (cf. the uploaded pdf), increasing the Transformer model size does not meaningfully improve performance on regular languages; the best values remain those originally used (`num_decoder_layers = 7, num_heads = 5`). For non-regular languages, the Transformer already outperformed the other architectures.

### **Expanding the discussion:**
While reviewers appreciated our discussion on the impact of our work, they suggested expanding it in the Discussion section. We summarise it here.

While other OOD generalisation types were examined in the literature, this is the first work studying rule extrapolation. This novel concept has the potential to impact LLM research both on conceptual and practical levels:
- General compositional generalization notions examine whether from learning multiple concepts/rules separately, the model can understand the composition of the concepts/intersection of the rules. However, in rule extrapolation we measure the reverse direction. Importantly, this notion of compositionality is less straightforward than the generally considered direction.
- Natural languages are compositional, thus, we expect this property in LLMs if they ought to model these languages well. Therefore, studying rule extrapolation can help us better understand LLMs’ inner workings.
- Rule extrapolation allows for studying compositional generalisation ability easily on a variety of datasets, such as formal or programming languages. Therefore rule extrapolation has the potential to become an established benchmark task for evaluating current and future LM architectures.

---

> ### Author Response · Authors · 2024-08-12
> **xLSTM results for the new context-sensitive language**
>
> We extend the results plotted in Figure 6 (b) in the uploaded pdf and provide the performance of the xLSTM on the new, context-sensitive Dyck language. We conclude that the xLSTM has similar performance to Mamba
>
> | Model                 | Test loss                 | ID R1                     | ID R2                     | OOD R1                    | OOD R2 $\tiny completion$ |
> |--------------------------------|------------------------------------|------------------------------------|------------------------------------|------------------------------------|------------------------------------|
> | Linear | $4.013\scriptscriptstyle\pm 0.254$ | $0.000\scriptscriptstyle\pm 0.000$ | $0.000\scriptscriptstyle\pm 0.000$ | $0.000\scriptscriptstyle\pm 0.000$ | $1.000\scriptscriptstyle\pm 0.000$ |
> | LSTM           | $0.645\scriptscriptstyle\pm 0.019$ | $0.981\scriptscriptstyle\pm 0.042$ | $0.956\scriptscriptstyle\pm 0.061$ | $1.000\scriptscriptstyle\pm 0.000$ | $0.894\scriptscriptstyle\pm 0.165$ |
> | Mamba          | $0.675\scriptscriptstyle\pm 0.018$ | $0.745\scriptscriptstyle\pm 0.070$ | $0.807\scriptscriptstyle\pm 0.185$ | $0.684\scriptscriptstyle\pm 0.159$ | $0.810\scriptscriptstyle\pm 0.212$ |
> | Transformer   | $0.640\scriptscriptstyle\pm 0.016$ | $1.000\scriptscriptstyle\pm 0.000$ | $1.000\scriptscriptstyle\pm 0.000$ | $0.980\scriptscriptstyle\pm 0.045$ | $0.973\scriptscriptstyle\pm 0.044$ |
> | xLSTM          | $0.671\scriptscriptstyle\pm 0.021$ | $0.791\scriptscriptstyle\pm 0.179$ | $0.765\scriptscriptstyle\pm 0.155$ | $0.767\scriptscriptstyle\pm 0.158$ | $0.715\scriptscriptstyle\pm 0.121$ |

---

### Decision · Program_Chairs · 2024-09-25

**Decision:**

Accept (spotlight)

**Comment:**

This paper proposes training models on data generated according to multiple rules, and evaluating models on generalization problems that involve extrapolating some of those rules to situations where others are violated. This is an interesting approach to generalization which differs somewhat from most prior OOD evaluations (even others on formal languages). The authors then systematically test different classes of models for extrapolation under several sets of rules, and find an interesting pattern of results. The reviewers generally agree that the paper is a useful contribution, especially after revision, and I concur.

I do think, however, that the work could situate itself better within the prior literature —  some prior papers have considered pitting structural rules against superficial ones in a way that resembles the rule extrapolation setting, e.g. https://aclanthology.org/2023.acl-short.38/ even if it does not explain it particularly clearly (see Fig. 1 caption).  Likewise, the prior Chomsky Hierarchy work (https://arxiv.org/abs/2207.02098) that came up in the discussion period seems worth discussing as well, even though it does not use the same extrapolation approach it is clearly a related space of issues. These are simply examples; I’d encourage the authors to do a more thorough literature review of related topics for the camera ready verison. These works do not impinge on the novelty of the current work, but discussing their relation would help to situate the reader and allow them to understand the current contribution better.